# Using *Medicago sativa* L. Callus Cell Extract for the Synthesis of Gold and Silver Nanoparticles

**DOI:** 10.3390/ijms262110772

**Published:** 2025-11-05

**Authors:** Inese Kokina, Ilona Plaksenkova, Lauris Jankovskis, Marija Jermaļonoka, Patryk Krzemiński, Aleksandra Mošenoka, Agnieszka Ostrowska, Renata Galek, Eriks Sledevskis, Marina Krasovska, Ligita Mežaraupe, Barbara Nasiłowska, Wojciech Skrzeczanowski, Maciej Chrunik, Marta Kutwin

**Affiliations:** 1Laboratory of Genomics and Biotechnology, Department of Technology, Institute of Life Sciences and Technology, Daugavpils University, Parādes Str. 1A, LV-5401 Daugavpils, Latvialaurisjankovskis@gmail.com (L.J.);; 2Department of Nanobiotechnology, Institute of Biology, Warsaw University of Life Science—SGGW, Jana Ciszewskiego Str. 8, 02-777 Warsaw, Polandagnieszka_ostrowska@sggw.edu.pl (A.O.); marta_kutwin@sggw.edu.pl (M.K.); 3Department of Genetics, Plant Breeding and Seed Science, Wroclaw University of Environmental and Life Sciences, Grunwaldzki Sq. 24A, 50-363 Wroclaw, Poland; renata.galek@upwr.edu.pl; 4Department of Technology, Institute of Life Sciences and Technology, G. Liberts’ Innovative Microscopy Centre, Daugavpils University, Parādes Str. 1A, LV-5401 Daugavpils, Latvia; 5Department of Ecology, Institute of Life Sciences and Technology, Daugavpils University, LV-5401 Daugavpils, Latvia; 6Institute of Optoelectronics, Military University of Technology, gen. S. Kaliskiego 2, 00-908 Warsaw, Poland; barbara.nasilowska@wat.edu.pl (B.N.); wojciech.skrzeczanowski@wat.edu.pl (W.S.); 7Institute of Applied Physics, Military University of Technology, gen. S. Kaliskiego 2, 00-908 Warsaw, Poland; maciej.chrunik@wat.edu.pl

**Keywords:** nanoparticles, alfalfa callus cells, reducing agents, extract, imaging

## Abstract

Gold (Au) and silver (Ag) nanoparticles (NPs) are used for drug transport and plant protection due to their insoluble nature and unique properties. To produce health-friendly NPs, toxic solvents should be replaced with plant-based synthesis. Plants, such as alfalfa (*Medicago sativa* L.), release biomolecules that reduce metal ions and form nanoclusters without free radicals, showing anti-inflammatory and antioxidant properties. In this study, callus cultures of two *M. sativa* genotypes, ‘Kometa’ and ‘La Bella Campagnola’, were exposed to two precursors (AgNO_3_ and HAuCl_4_) for 24 and 48 h to assess the feasibility of biological NP synthesis. Spectrophotometry showed significant (*p* ≤ 0.05) changes in light absorbance compared with the control. Dynamic light scattering and zeta potential measurements indicated a change in the composition of the liquid compared with the control. To improve image quality and obtain more accurate data, transmission electron microscopy (TEM) analysis was repeated, confirming the presence of quasi-spherical nanoparticles with diameters in the range of 5–25 nm for both AuNPs and AgNPs in the callus culture extracts of both genotypes. Nanoparticle Tracking Analysis demonstrated that the AgNPs and AuNPs from both genotypes displayed polydisperse size distributions, with a mean particle size ranging from 220 to 243 nm. Elemental analysis provided clear evidence that Ag and Au were present only in treated samples, confirming effective NP biosynthesis and excluding contamination. X-ray diffraction (XRD) analysis was performed to characterise the crystalline structure; however, due to the very small particle size (5–25 nm), no clear diffraction patterns could be obtained, as nanocrystals below ~20–30 nm typically produce signals below the detection limit of standard XRD instrumentation. The novelty of this research is the cost-effective, rapid biosynthesis of quasi-spherical AuNPs and AgNPs with diverse sizes and enhanced properties, making them more eco-friendly, less toxic, and suitable for antibacterial and anticancer studies.

## 1. Introduction

Artificially synthesised nanoparticles (NPs) induce abiotic stress [1] and cytotoxic, genotoxic effects [2]. Silver nanoparticles (AgNPs) and gold nanoparticles (AuNPs) can impair plant growth by accelerating the formation of reactive oxygen species. Stress responses alter biochemical parameters and reduce photosynthesis efficiency [3,4,5], resulting in irregular morphological changes that reduce plant viability [1]. To reduce NP (nanoparticle) toxicity and improve environmental safety, it is essential to thoroughly study synthesis methods. Synthesis must minimise NP toxicity while retaining beneficial properties.

AgNPs and AuNPs are widely used in medicine, the food industry, and cosmetics [6,7,8,9]. They show a wide range of antimicrobial and therapeutic activities [6,7,8,9,10,11,12]. These qualities enable both AgNPs and AuNPs to be used in imaging and biomedical devices, as well as diagnostics [10,11,12]. Their applications include dentistry wound dressings, Ag-coated catheters, ventricular drainage systems, and orthopaedic infection control [13,14].

AgNPs and AuNPs can be synthesised using physical (top-down) and chemical (bottom-up) methods. In the top-down approach [8], NPs are derived from bulk materials through evaporation–condensation, laser ablation, and gamma irradiation. In contrast, the bottom-up approach involves the formation of NPs from molecular components, typically through the nucleation and growth of precursor salts. This method encompasses a variety of techniques, including microemulsions and photoreduction. However, these conventional methods often involve toxic reagents and are time-consuming [3,6,7,8]. This increases demand for safer synthesis methods that avoid toxic chemicals and maintain NP functionality [15,16,17,18]. Biological synthesis is increasingly being used to obtain less toxic NPs with improved chemical and physical properties [6,9]. This method involves breaking down the precursor to the nanoscale using bacteria, algae, fungi, yeasts, or plants [19]. Plant-based synthesis is particularly favoured, as plant extracts contain a high concentration of biomolecules [20,21,22], which act as natural reducing agents [23,24].

Plants can effectively decompose metals by absorbing them from the soil into their cells, where they are broken down in vivo and stored [25]. NPs may accumulate in organisms and enter the food chain [26]. The most commonly used and best-controlled biological method for obtaining NPs involves using plant cell extracts [23,24,27,28,29,30], which can be derived from the leaves, stems, fruits, flowers, or roots [31,32,33,34]. Callus cultures also serve as a useful source of extracts due to their high biomolecule content [24,27,30,32,35]. Plant tissue is effective as a model system for plant physiology studies [31]. Some researchers have already utilised plant callus cultures for the biological production of NPs [24,27,28,29,30]. In this study, callus cultures from two alfalfa (*Medicago sativa* L.) genotypes were used for AuNP and AgNP biosynthesis for the first time. Previous research has explored the biological synthesis of AgNPs and AuNPs using seedlings of the alfalfa genotypes ‘Kometa’ and ‘La Bella Campagnola’, but similar studies on callus cultures of these genotypes have yet to be performed [25].

*Medicago sativa* contains a diverse array of biomolecules, including glucose, fructose, phenols, citric acid, ascorbic acid, proteins, flavonoids, amino acids, polyphenols, and reducing sugars [31,36]. These compounds are present in even higher concentrations in alfalfa callus cultures [37,38,39]. Consequently, AgNPs and AuNPs can be synthesised more efficiently from alfalfa callus culture extracts than from 2-week-old seedlings [40,41]. Plant extracts, particularly those from alfalfa, reduce metal ions (Me^n+^) to their elemental form (Me^0^) and stabilise the solution. Polyphenols and flavonoids are the primary metal ion-reducing compounds responsible for NP formation [42,43,44]. Phenolic compounds donate electrons through hydroxyl groups, reducing Ag^+^ and Au^3+^ ions to nanoparticles, while their aromatic rings stabilise growing NP nuclei [10,43,44]. Flavonoids also participate in both reduction and capping processes, preventing NP aggregation [42].

Successful biological NP synthesis requires precise preparation of the extract and the correct precursor concentration during the synthesis process [45,46]. However, monitoring NP formation in plant extracts can be challenging due to weak visual indicators during synthesis [47,48]. Rapid and cost-effective analytical methods for characterising plant extract composition remain limited [49]. To address this issue, the present study incorporated advanced microscopy techniques to provide high-quality visualisation of callus culture extracts, clearly confirming the formation of AgNPs and AuNPs in alfalfa extracts. Previous studies confirm successful NP formation using plant-based synthesis approaches [46]. This highlights the potential of biological NP synthesis methods and the need for continued research [15,16,17].

## 2. Results

### 2.1. Medicago sativa Seedling and Callus Culture Cultivation

*Medicago sativa* ‘Kometa’ and ‘La Bella Campagnola’ seedlings were successfully grown. The seedlings measured 7–9 cm in length after 14 days of germination. They were of high quality and suitable for obtaining callus cultures. Healthy ‘Kometa’ and ‘La Bella Campagnola’ callus cultures were used for the subsequent experiments.

### 2.2. Obtaining M. sativa Callus Culture Extracts for AgNP and AuNP Biosynthesis

Despite the relatively small biomass of the alfalfa callus cultures, plant green mass suspensions were successfully obtained from both experimental groups for spectrophotometric analysis. When evaluating the external appearance of the suspensions, there was a drastic colour change from green to red in the samples exposed to HAuCl_4_ for 24 h. After exposure for 48 h, the samples exhibited a darker red colour. This colour change occurred due to localised surface plasmon resonance (SPR), a phenomenon in which NP formation and growth influence their absorption spectrum. AuNPs with a size ranging from 10 to 100 nm absorb light in the visible spectrum (520–550 nm), producing an intense red or violet colour [50,51]. Conversely, there was no significant colour change in the samples treated with AgNO_3_. The SPR absorbance peak of AgNPs typically occurs in the 350–450 nm range, which lies within the UV–Vis transition region. Because the human eye is less sensitive to this spectral range, any colour change in the solution may be weak or even imperceptible [52]. There were no precipitates, deposits, or particles of other materials in these samples.

### 2.3. Determination of Light Absorbance Intensity

A NanoDrop 1000 spectrophotometer was used to measure light absorbance in the suspension solution at 350 and 470 nm for AgNO_3_-treated samples and at 450 and 650 nm for HAuCl_4_-treated samples. To ensure the accuracy of the absorbance readings, each sample (*n* = 10 per genotype, treatment, and exposure time) was analysed in triplicate, resulting in 30 measurements per genotype. Figure 1 presents the average absorbance change at different wavelengths after exposure to the precursor for 24 h compared with the control samples. For both genotypes, absorbance at 450 nm increased significantly after treatment with HAuCl_4_. In the ‘La Bella Campagnola’ samples, absorbance at 650 nm also increased significantly. Absorbance at 350 nm increased significantly in the ‘Kometa’ and ‘La Bella Campagnola’ AgNO_3_-treated samples. In the remaining samples, absorbance at 470 nm was not significantly different compared with the control samples, although there was a slight, nonsignificant decrease in absorbance for both genotypes treated with AgNO_3_.

Figure 2 presents the mean absorbance change at different wavelengths after exposure to the precursors for 48 h compared with the control samples. Absorbance at 450 nm increased significantly for the ‘Kometa’ HAuCl_4_-treated samples, and absorbance at 350 nm increased significantly for the ‘Kometa’ and ‘La Bella Campagnola’ AgNO_3_-treated samples. There was a nonsignificant increase in absorbance at 450 and 650 nm in the ‘La Bella Campagnola’ HAuCl_4_-treated samples. In the remaining samples, absorbance at 470 nm did not show a significant change compared with the control, except for the ‘La Bella Campagnola’ AgNO_3_-treated samples, which showed a nonsignificant decrease.

### 2.4. Determination of Light Absorbance

A two-beam Shimadzu UV-2550PC UV–Vis spectrophotometer was used to measure absorbance at wavelengths ranging from 340 to 900 nm. The absorbance data for each *M. sativa* genotype, based on the precursor type and exposure duration, are presented in Table 1 and Figure 3.

For the ‘Kometa’ HAuCl_4_-treated samples (Figure 3a,b), there was a distinct SPR peak at 534 nm after exposure for 24 and 48 h, confirming the formation of AuNPs. Over time, the peak intensity increased, suggesting continued NP formation, potential growth in size, or increased aggregation. Additionally, there was strong absorbance in the UV region at 286 nm after exposure for 24 and 48 h, which may be attributed to small Au clusters, ligand-to-metal charge transfer interactions, or biomolecule–NP interactions [53]. For the ‘Kometa’ AgNO_3_-treated samples, there was a prominent peak at 286 nm after exposure for 24 and 48 h; however, the characteristic AgNP SPR band in the 400–450 nm range was absent, suggesting that Ag primarily remained in an ionic or small-cluster form, rather than forming stable, well-defined plasmonic NPs [51]. The slight increase in absorbance intensity over time indicates an ongoing interaction between Ag species and biomolecules but without significant NP growth or aggregation. The control sample containing only *M. sativa* extract exhibited consistently low absorbance at both time points, confirming that the spectral features observed in the metal-treated samples were due to NP formation rather than intrinsic absorption from alfalfa components.

The ‘La Bella Campagnola’ HAuCl_4_-treated samples (Figure 3c,d) presented a distinct SPR peak at 286 nm for both exposure times, confirming the formation of AuNPs. The peak intensity increased over time, suggesting continued NP formation, a potential size increase, or increased aggregation. In addition, there was strong absorbance in the UV region at 534 nm after exposure for both 24 and 48 h, which could be attributed to small Au clusters, ligand-to-metal charge transfer interactions, or biomolecule–NP interactions. Similarly to the ‘Kometa’ samples, the ‘La Bella Campagnola’ AgNO_3_-treated samples exhibited a significant absorbance peak at 286 nm after exposure for 24 and 48 h but lacked the characteristic SPR band of AgNPs in the 400–450 nm range. This suggests that Ag remains primarily in the form of ions or small clusters, rather than forming stable, well-defined plasmonic NPs. The slight increase in the absorbance intensity over time indicates ongoing interactions between Ag species and biomolecules but without significant NP growth or aggregation. The ‘La Bella Campagnola’ control sample containing only *M. sativa* extract also exhibited consistently low absorbance at both time points, confirming that the spectral features observed in the metal-treated samples were due to NP formation and not the intrinsic absorption of alfalfa components.

### 2.5. Transmission Electron Microscopic Visualisation of M. sativa Callus Culture Extract

Transmission electron microscopy was used to visualise the callus culture extracts from both genotypes. The micrographs are shown in Figure 4.

In the ‘Kometa’ (Figure 4a) and ‘La Bella Campagnola’ (Figure 4b) negative control samples at 1 µm scale, fragments of the *M. sativa* callus culture extract were visible with layered structures piled sequentially in several layers of water. The largest, darkest structures that formed large black spots (circled in blue) were thought to be a particle agglomeration, likely being parts of the plant fragments disjoined from the plant. Other black dots were thought to represent fragments of the plant material.

In the callus culture extracts exposed to HAuCl_4_ (sample ‘Kometa’—Figure 4c and ‘La Bella Campagnola’—Figure 4d) and AgNO_3_ (sample ‘Kometa’—Figure 4e and ‘La Bella Campagnolla’—Figure 4f) at 200 nm scale, fragments of *M. sativa* callus culture extract were again observed with layered structures piled up sequentially in several layers of water. All samples did exhibit spherical structures <100 nm in size, which were identified as NPs (circled in yellow/red arrows). Dynamic light scattering (DLS) analysis confirmed that all samples involved metal particles with <100 nm size. In the *M. sativa* callus culture samples, ‘Kometa’ samples treated under HAuCl_4_ treatment demonstrated a size range of approximately 10–60 nm with average diameters of around 25–40 nm (Figure 4c) and displayed AgNP particle sizes of around 10–50 nm with average sizes approximately around 25–35 nm (40–50 nm) (Figure 4e). Under the treatment with HAuCl_4_, the ‘La Bella Campagnola’ genotype had AuNPs which resulted in approximate sizes in the range 10–70 nm with average sizes of around 25–40 nm (Figure 4d), and also displayed AgNP particle sizes of around 10–50 nm with an average around 25–35 nm (Figure 4f).

In the distribution of HAuCl_4_ tested in both genotypes, it appeared to be distributed more uniformly than AgNO_3_; nevertheless this data was not statistically confirmed. Sample ‘La Bella Campagnola’ treated with HAuCl_4_ (Figure 4d) was the sample with the highest density of particles, with approximately 200–250 particles per 1–2 µm^2^, while sample ‘Kometa’ treated with HAuCl_4_ treatment was at 100–150 particles per 1–2 µm^2^ (Figure 4c); moreover, treatment with AgNO_3_ particles in the genotypes yielded relatively minimal density counts of particles for ‘Kometa’, with approximately 30–40 particles per 0.5–1 µm^2^ (Figure 4e). AgNO_3_ treatment of sample ‘La Bella Campagnola’ yielded minimal particle counts of 20–30 per 0.5–1 µm^2^ (Figure 4f).

The largest, darkest structures that formed large black spots (circled in blue) were considered to be agglomerated particles, most likely disjoined plant pieces. However, in these samples, we had noticed more agglomerated structures that seemingly obtain varying degrees of rigidity which may have been residual matter of NP synthesis or possibly NPs that could have agglomerated to form large-mass clusters. For treatment of ‘Kometa’ by HAuCl_4_ (Figure 4c), we estimate produced agglomerates of approximately 80–200 nm with approximately 30–40% of the particles agglomerated, compared to AgNO_3_ treatment (Figure 4e), yielded agglomerates of 60–120 nm, with approximately 40–50% particles agglomerated. For the ‘La Bella Campagnola’ reactants, reacted with either the AuCl_4_ (Figure 4d) or AgNO_3_ (Figure 4f), there were agglomerates of 60–200 nm sized particles, approximately the same as the estimates, as maybe around 20–30% of the particles agglomerated. However, AgNPs appeared to aggregate more thoroughly than AuNPs in the genotypes, perhaps indicating a larger differentiation of colloidal stability.

There were smaller particles that did not exhibit any agglomeration but are still circled in green. Based on their size, they were not considered NPs, but they possibly remained as fragments of residual matter of synthesis—literally parts of the precursor still in the decomposition process. The overall background was shown to be fairly clean with very minor graininess, likely to be artefacts left over from the substrate (carbon grids), pointing to high-quality synthesis with few organic or inorganic residues remaining.

### 2.6. DLS and the Zeta Potential

DLS analysis was performed to determine the Z-average and zeta potential for the ‘Kometa’ and ‘La Bella Campagnola’ control, HAuCl_4_-treated, and AgNO_3_-treated samples. Additionally, a graph of the particle size distribution by intensity was obtained. There were three peaks of different sizes in all samples treated with the precursors, whereas the control samples showed two peaks of different sizes, suggesting the presence of different particle populations. Table 2 provides a summary of the physicochemical characterisation data for biosynthesised AuNPs and AgNPs.

The Z-average obtained from DLS describes the average hydrodynamic diameter and the degree of dispersion of the particles in the samples. The ‘Kometa’ and ‘La Bella Campagnola’ control samples exhibited relatively large particle sizes (260 and 552 nm, respectively). The ‘Kometa’ and ‘La Bella Campagnola’ HAuCl_4_-treated samples showed high polydispersity in size (1074 ± 10 nm), suggesting a heterogeneous size distribution. The ‘Kometa’ and ‘La Bella Campagnola’ AgNO_3_-treated samples exhibited smaller average sizes (324–322 nm), indicating a relatively more uniform distribution.

The PDI indicates the uniformity of particle size distribution. It describes how homogeneous or heterogeneous the particle size distribution is in a solution. The PDI exceeded 0.4 for almost all samples. A PDI > 0.3 indicates that the sample contains a heterogeneous (polydisperse) system with a wide range of particle sizes and potential aggregations, which may affect stability [54,55,56]. The PDI for both genotypes treated with HAuCl_4_ or AgNO_3_ increased significantly compared with the control, indicating that after exposure to the precursor and potential NP synthesis, there was an increase in the particle size diversity and the number of particles.

The zeta potential indicates the surface electric charge of particles and characterises their stability in a colloidal solution. It is the electric potential formed between a liquid layer that closely adheres to the particle surface and the surrounding liquid [9]. The ‘Kometa’ control sample was the most unstable, with a zeta potential of −10 mV. A solution with a zeta potential of at least ±13 mV is considered stable and is referred to as having moderate colloidal stability. The ‘Kometa’ samples treated with either precursor showed a significant decrease in zeta potential compared with the control sample (HAuCl_4_: −16 mV, AgNO_3_: −15 mV), indicating an increase in charge stability in these samples. There was a similar trend for the ‘La Bella Campagnola’ samples. The control sample showed moderate colloidal stability (−13 mV), but the zeta potential of the experimental samples decreased significantly (HAuCl_4_: −17 mV, AgNO_3_: −16 mV). This suggests that following possible NP synthesis, charge stability increased. Notably, the zeta potential, PDI, and Z-average did not differ significantly between the ‘Kometa’ and ‘La Bella Campagnola’ samples.

### 2.7. NTA

The NTA (Nanoparticle Tracking Analysis) results using the NanoSight Pro system revealed that both AgNPs and AuNPs in the ‘Kometa’ and ‘La Bella Campagnola’ samples exhibited polydisperse size distributions, with an average particle size of approximately 220 to 243 nm (Table 3).

In the ‘Kometa’ AgNO_3_-treated sample, the modal particle diameter was 128 nm, with a mean diameter of 220 nm and a D90 of 385 nm. The particle concentration was 7 × 10^8^ particles/mL. The ‘Kometa’ HAuCl_4_-treated sample exhibited the same mode (128 nm), a mean of 243 nm, a D90 of 440 nm, and a particle concentration of 9 × 10^8^ particles/mL. The ‘Kometa’ control sample showed a mode of 253 nm, a mean of 284 nm, and a D90 of 492 nm. In the ‘La Bella Campagnola’ AgNO_3_-treated sample, the mode was 143 nm, the mean was 219 nm, and the D90 was 384 nm, with a particle concentration of 4 × 10^8^ particles/mL. The ‘La Bella Campagnola’ HAuCl_4_-treated sample had a mode of 173 nm, a mean of 242 nm, a D90 of 449 nm, and a concentration of 4 × 10^8^ particles/mL. The ‘La Bella Campagnola’ control sample exhibited a mode of 143 nm and a mean size of 223 nm.

The NTA spectra shown in Figure 5 were used to compare different reaction conditions for the synthesis of AgNPs and AuNPs. Regarding the samples treated with AgNO_3_ (Figure 5A), the AgNPs synthesised with the ‘La Bella Campagnola’ callus culture extract exhibited a narrower size distribution and a smaller modal diameter, indicating a more uniform particle population. For the samples treated with HAuCl_4_ (Figure 5B), the AuNPs produced with the ‘Kometa’ callus culture extract showed a broader size distribution and the presence of larger particles or aggregates, suggesting less controlled reaction conditions or a tendency for aggregation.

### 2.8. Elemental Analysis of Biogenic NPs Using EDS

EDS analysis was performed on biogenic nanostructures in the *M. sativa* ‘Kometa’ and ‘La Bella Campagnola’ callus culture extracts to validate the successful formation of metallic NPs and to assess their elemental composition. The results are summarised in Table 4. The EDS spectra revealed clear elemental signatures corresponding to the metallic core of the NPs. For the ‘Kometa’-derived AgNPs, there was a high Ag content, reaching approximately 39 wt%, with minimal contributions from silicon (Si), phosphorus (P), potassium (K), calcium (Ca), and trace elements commonly associated with biological matrices. Similarly, the ‘Kometa’-derived AuNPs showed a distinct Au peak corresponding to a mean content of ~25 wt%, accompanied by residual plant-associated elements (e.g., carbon [C], chlorine [Cl], K, and Ca). The control ‘Kometa’ sample, devoid of added metal precursors, contained predominantly C, oxygen (O), and Si, consistent with the composition of the dried phytomatrix. Comparable findings were obtained for the ‘La Bella Campagnola’-derived NPs. For the AgNPs, Ag accounted for ~41 wt%, confirming robust reduction and stabilisation of Ag ions by the phytochemicals present in the extract. The Au content in the AuNPs reached ~26 wt%, supporting efficient bioreduction of Au^3+^. In both cases, typical elements from the organic matrix (e.g., C, Si, K, and Cl) were also observed. The control ‘La Bella Campagnola’ sample showed an elemental profile analogous to that of the ‘Kometa’ control sample, lacking any signal from noble metals.

### 2.9. LIBS of the Tested NPs

The LIBS (laser-induced breakdown spectroscopy) spectra were obtained for the *M. sativa* ‘Kometa’ and ‘La Bella Campagnola’ control, AgNO_3_-treated, and HAuCl_4_-treated callus culture extracts and confirmed the presence of AgNPs and AuNPs (Figure 6). For AgNPs, the LIBS spectra clearly demonstrated characteristic emission peaks corresponding to neutral Ag atoms (Ag I) at 328.069 and 338.289 nm. The observed peaks were pronounced in the ‘Kometa’ and ‘La Bella Campagnola’ AgNO_3_-treated samples, confirming successful incorporation of Ag ions into the biogenic nanostructures. The intensity of the Ag peaks was notably absent in the control samples, supporting the view that no background Ag signal originated from the plant matrix alone. Similarly, the presence of Au in the ‘Kometa’ and ‘La Bella Campagnola’ HAuCl_4_-treated samples was confirmed by the distinct Au I emission line at 267.595 nm. The Au signal was consistent for both *M. sativa* genotypes, and its absence in the control samples further validated the specificity of NP formation and ion binding.

### 2.10. X-Ray Diffraction (XRD) Analysis

Due to the weak diffraction signal revealed by both types of nanoparticles, a single X-ray diffraction measurement was not processed by means of background subtraction and λK_α2_ component removal before the actual phase analysis (one can observe in Figure 7A,B a raw signal acquired from the diffractometer). An attempt to obtain any XRD signal required the preparation of relatively thick layered surfaces of the studied nanostructures on a glass substrate. In the case of phase analysis of Ag nanoparticles (regardless of the ‘Kometa’ and ‘LaBella Campagnola’ sources), characteristic reflections labelled as (**◊**) stand for cubic Ag (plane (hkl) indices in parentheses), as shown in Figure 7A, most strongly from the (111) plane. However, these reflections are very weak and have large half-widths. Additionally, in both cases, the presence of the AgNO_3_ precursor (●) and some unidentified phases (*****) is noticeable (impossible to recognise based on the knowledge of elemental analysis from the subsequent chapters of this work and using the COD crystallographic database supporting the program). Focusing solely on the peak intensity from the (111) plane, one can suggest that the level of crystallinity (or concentration) of Ag nanoparticles is higher in the case of the ‘LaBella Campagnola’ source. However, the background/noise-to-signal ratio is still quite high in both sources, which made this analysis difficult. The presence of metallic Ag structures is confirmed, but both sample sources indicate inhomogeneity. The situation is similar for (**▪**) Au nanoparticles (Figure 7B), but the diffraction signal from the metallic nanoparticles themselves is slightly improved compared to Ag, although the situation is a bit better for the ‘Kometa’ source when compared to the ‘LaBella Campagnola’ one. Unidentified phases (*****) also appear simultaneously for these nanoparticles, regardless of the source, and are of the same type in Kometa and ‘LaBella Campagnola’. Also, in this case, we are dealing with inhomogeneous samples. The probable reason for both (Ag and Au sources) inhomogeneities is residuals of metallic precursors or other organic impurities in the suspensions. Because the Ag and Au structures are identical in terms of space group and differ only slightly in lattice constants (of hundredths of an angstrom), the diffraction positions of their characteristic lattice planes in the XRD patterns differ only slightly.

### 2.11. Comparison of the Results After Exposure to the Precursor for 24 or 48 h

A one-way ANOVA was conducted to evaluate potential differences in light absorbance between the samples exposed to the precursor for 24 or 48 h. Across all samples, there were no significant differences in light absorbance intensity between the time points. There was a significant difference between the control and experimental (i.e., AgNO_3_-treated and HAuCl_4_-treated) samples, but no significant differences among the experimental samples themselves.

## 3. Discussion

Although visualising metal NPs in plant tissues requires advanced analytical techniques that demand more extensive sample preparation [20,57,58,59,60], this study confirmed that *M. sativa* callus culture extracts were capable of biosynthesising AuNPs and AgNPs from AgNO_3_ and HAuCl_4_, respectively, and transmission electron microscopy successfully visualised their presence in callus cells. To improve image quality and obtain more accurate data, TEM analysis was repeated. Small, quasi-spherical NPs were observed, with both AuNPs and AgNPs having sizes in the range of 5–25 nm, depending on the genotype [32,33,36]. Furthermore, NPs smaller than 4–15 nm can be obtained in the extracts of other plants, such as *Apium graveolens* L. leaves [61]. A recent study used confocal microscopy to demonstrate that both HAuCl_4_ and AgNO_3_ can penetrate ‘Kometa’ and ‘La Bella Campagnola’ cells [25].

The results of this study revealed that the sample, when suspended in a hydrocolloid solution, consisted of multiple layers of cells and particulate matter. Transmission electron microscopy identified three distinct particle groups based on size—large agglomerates, medium-sized synthesis residues, and small NPs—with only the smallest group falling within the nanoscale range. Sample multilayering is typical for plant-based NP synthesis due to residual biomolecules and incomplete homogenisation of plant tissue, which affects NP visibility and purity [62,63]. Additional purification methods, such as dialysis, can improve NP isolation and image clarity [33,36,57], but were not applied in this study in order to preserve the natural interaction between biomolecules and nanoparticles during synthesis. This approach is consistent with previous reports [64,65,66], in which samples were purified during synthesis but no extra post-synthesis purification step was applied prior to TEM analysis. Quasi-spherical NPs were predominant in this study, which is commonly reported in green synthesis due to biomolecule-controlled nucleation [58,59,60]. TEM imaging was repeated to ensure reproducibility of NP morphology and confirm the presence of both AgNPs and AuNPs across independent preparations.

The zeta potential data also indicated moderate colloidal stability, with all samples exceeding the ±13 mV threshold. The most negative recorded zeta potential was −17.22 mV, suggesting relative stability but a continued tendency towards agglomeration. Zeta potential values more negative than −30 mV typically indicate strong electrostatic repulsion and high colloidal stability, while values between −10 mV and −30 mV correspond to moderate stability [29,51,63]. Such values suggest that the phytochemical components of the *Medicago sativa* callus extract, including phenolic acids, flavonoids, and proteins, adsorbed onto the nanoparticle surfaces, providing both electrostatic and steric stabilisation. The relatively high PDI values (>0.4) observed for all samples indicate a broad particle size distribution and the coexistence of small and aggregated nanoparticles, which is typical for biologically synthesised systems where multiple biomolecules act as reducing and capping agents. Similar trends have been reported for plant-mediated nanoparticle synthesis [67].

UV–Vis spectrophotometry is recognised as a reliable, convenient, and effective technique for the primary characterisation of synthesised AgNPs. Additionally, it is widely used to monitor NP synthesis and stability [68,69]. In the present study, UV–Vis spectrophotometry confirmed the successful synthesis of both AgNPs and AuNPs. There was a significant increase in absorbance at 450 nm after exposure to HAuCl_4_—particularly in the ‘Kometa’ samples—which indicates efficient AuNP formation. Similarly, increased absorbance at 350 nm following exposure to AgNO_3_ confirmed AgNP synthesis. Most NP formation occurred within the first 24 h of incubation, which is typical for green synthesis where flavonoids rapidly reduce metal ions [68,70,71]. However, no distinct SPR peak at ~420 nm was observed for AgNPs in this study. This may be due to overlapping absorption from plant biomolecules, NP aggregation, or the formation of very small AgNP clusters (<10 nm) that do not exhibit a defined plasmon resonance [53,63,69]. The presence of AgNPs was still confirmed by TEM and NTA, suggesting that UV–Vis alone may not be conclusive in complex biological matrices. The plant extract, directly involved in the synthesis process, forms a coating layer composed of biomolecules on the NP surface. These biomolecules, mainly phenolic compounds, proteins, and flavonoids, act as both reducing and capping agents, controlling NP nucleation and growth and preventing aggregation [72,73]. This capping layer contributes to the stability and size distribution of AgNPs and AuNPs synthesised in this study.

The AgNPs showed absorption at an atypical wavelength, 286 nm, which can be attributed to a biomolecular film coating—particularly protein—on the NP surface. A previous study reported a similar peak at 290 nm for AgNPs, attributed to NP–protein interactions [73]. Another study noted a peak at 280 nm for both AgNPs and AuNPs, which increased with the synthesis time due to NP–protein interactions [74]. In this study, both AgNPs and AuNPs showed a similar absorption peak at 286 nm, suggesting strong interaction between phytochemicals from the callus extract and the NP surface. Notably, the initial HAuCl_4_ concentration was one-fifth that of the initial AgNO_3_ concentration. Precursor concentration strongly influences NP nucleation and growth rates [68]. Excessive precursor and biomolecule concentrations in the plant extract can lead to incomplete AgNP formation and aggregation, impairing NP quality. The high initial precursor concentration in the current study likely affected AgNP formation and the UV–Vis results [69]. The elevated AgNO_3_ concentration probably delayed particle formation and promoted NP cluster formation, altering the absorption spectrum. These findings indicate that biomolecules present in the callus extract, such as phenolics and proteins, may have formed a capping layer on the NPs, shifting absorption towards lower wavelengths. Furthermore, the size of AgNPs synthesised via the green method increases as the incubation period (and thus reaction time) increases [75], an effect attributed to colloidal AgNP agglomeration [75]. The NP shape also significantly influences optical properties. Irregularly shaped AgNPs exhibit multiple plasmon resonances, depending on their proportions [69].

A slight decrease in absorbance at 470 nm after AgNO_3_ treatment may indicate early stages of particle stabilisation and cluster formation.

The particle sizes observed by TEM in this study (5–25 nm) were smaller than those measured by NTA, which also detected larger particle populations (>100 nm). This discrepancy is common in green synthesis and can be explained by the fact that TEM measures only individual, dried particles, while NTA detects hydrodynamic diameters of particles in suspension, including biomolecule corona layers and nanoparticle agglomerates [53,63,76]. Moreover, TEM provides the core diameter of nanoparticles in a dry state, while DLS and NTA measure the hydrodynamic diameter of particles dispersed in solution, which includes the solvation layer, surface-bound biomolecules from the callus extract, and occasional aggregates. These factors account for the larger mean sizes observed in DLS/NTA relative to TEM, consistent with findings in similar green synthesis systems [77]. Overall, AuNP synthesis appeared more efficient in *M. sativa* ‘Kometa’, while *M. sativa* ‘La Bella Campagnola’ showed signs of retaining more precursor or biomolecular residues, possibly due to differences in biochemical composition. These type-dependent differences may be associated with variations in the concentration of reducing and capping agents such as phenolics, flavonoids, and proteins in the callus extracts. A higher content of these biomolecules in ‘Kometa’ could accelerate metal ion reduction and promote the formation of more uniform nanoparticles, while lower biomolecular activity in ‘La Bella Campagnola’ may result in slower nucleation and increased particle aggregation [31,36,37].

While the SPR peak typical for AgNPs (420 nm) was not clearly visible, this does not exclude AgNP formation; the presence of AgNPs was still supported by the transmission electron microscopy and NTA results. The AuNP absorbance peaks near 534 nm matched the typical SPR range (500–550 nm), supporting successful formation. Previous studies have reported similar trends, although with some variation in the peak position depending on the synthesis conditions and the plant extract composition [53,63]. Although AgNPs typically exhibit a characteristic SPR band at ~420 nm, this signal was not clearly visible in the present study. The absence or shift of the AgNP SPR band is commonly reported in green synthesis systems and is attributed to strong surface interactions between nanoparticles and phytochemicals. Phenolic compounds, proteins, and other metabolites may form a capping corona around AgNPs, altering the local dielectric environment and suppressing or shifting the plasmon resonance [53,63,69]. Additionally, very small AgNP nuclei (<10 nm) and early-stage clusters often lack a well-defined SPR band in complex biological matrices. These effects explain why AgNPs in this study exhibited a peak at ~286–290 nm rather than the typical plasmon range, despite their confirmed presence in TEM and NTA analyses.

These findings suggest that biosynthesis efficiency is influenced more by the genotype, precursor concentration, and biomolecular content than by exposure time.

UV–Vis analysis alone did not fully confirm AgNP and AuNP formation due to interference from plant-derived compounds. The AuNP spectra showed a clearly visible SPR band at approximately 530 nm. However, the AgNP spectra did not contain an SPR band at approximately 400–420 nm, which is characteristic of AgNPs. Instead, AgNPs exhibited a band at ~290 nm, which is commonly attributed to nanoparticle–biomolecule interactions. This shift in absorbance may result from biomolecular capping by proteins and phenolic compounds, which can mask the metallic plasmon resonance. TEM analysis nevertheless clearly confirmed AgNP and AuNP formation. Additional purification steps were not applied in order to maintain the original composition of the callus extract during analysis [78,79]. Nevertheless, NTA was also performed to verify that AuNPs, and especially AgNPs, were synthesised in the callus culture extracts.

The NTA results confirmed the successful biosynthesis of both AgNPs and AuNPs, particularly in the ‘Kometa’ AgNO_3_-treated sample. In this sample, the particles had a mode of 127.5 nm, a mean of 220 nm, and a D90 of 385 nm, indicating the presence of larger particles or some aggregation. The high particle concentration (7.41 × 10^8^ particles/mL) suggests a significant amount of NPs in the suspension. Similarly, the ‘Kometa’ HAuCl_4_-treated sample had the same mode (127.5 nm), a slightly higher mean (243 nm), and a broader D90 (440 nm), along with an even higher concentration (9.08 × 10^8^ particles/mL), supporting the formation of AuNPs as well. The ‘Kometa’ control sample showed much larger particles (mode of 252.5 nm, mean of 284 nm, and D90 of 492 nm), likely related to background particles from the matrix, not added NPs. The ‘La Bella Campagnola’ AgNO_3_-treated sample had a mode of 142.5 nm, a mean of 219 nm, and a D90 of 384 nm, with a particle concentration of 3.96 × 10^8^ particles/mL. These values also support AgNP formation, though to a slightly lesser extent than in the ‘Kometa’ AgNO_3_-treated sample. The ‘La Bella Campagnola’ HAuCl_4_-treated sample showed more variation in particle sizes (mode of 172.5 nm, mean of 242 nm, and D90 of 449 nm) and a concentration of 4.35 × 10^8^ particles/mL, suggesting some aggregation but still indicating AuNP formation. The ‘Kometa’ and ‘La Bella Campagnola’ control samples had size distributions that partly overlapped with the AgNO_3_- and HAuCl_4_-treated samples, showing that the base formulation contributes some background particles. However, the changes in size and higher particle concentrations in the AgNO_3_- and HAuCl_4_-treated samples confirm that NP biosynthesis did take place.

These results indicate that ‘Kometa’ generally produced a higher nanoparticle yield and narrower size distribution compared to ‘La Bella Campagnola’, suggesting greater reducing efficiency and faster nucleation in this genotype. The broader D90 values and higher mean diameters reflect the tendency of biosynthesised NPs to form agglomerates in plant extract media, which is common in green synthesis due to biomolecular interactions.

Single-particle tracking via NTA, as reported in the literature [79,80,81], detects only particulate matter above ~30 nm and does not generate signals for dissolved ions or precursors such as HAuCl_4_ or AgNO_3_. Therefore, the signals observed in our samples can be attributed exclusively to nanoparticles. To further support this interpretation, control measurements were performed with silver nitrate (AgNO_3_) and perchloric acid (HClO_4_) solutions. HClO_4_ was used solely as a comparative control to demonstrate that NTA detects only particulate matter undergoing Brownian motion and does not generate signals from molecular precursors or dissolved ions. Indeed, for AgNO_3_, we observed a significantly higher nanoparticle density compared to the control extract, confirming that the signal originates from particulate structures. In contrast, no particles were detected in the HClO_4_ sample, and no dot plot could be generated. These results further strengthen the conclusion that only nanoparticles contribute to the NTA signal.

These findings confirm that the measured particles were not artefacts caused by precursor residues but actual nanoparticles formed during biosynthesis in the callus extracts. NTA therefore provided independent quantitative validation of nanoparticle formation, complementing TEM and UV–Vis analyses.

EDS (energy-dispersive X-ray spectroscopy) and LIBS were performed to further verify the presence of AgNPs and AuNPs and to overcome the limitations observed with the UV–Vis spectra—particularly the lack of a distinct SPR band for AgNPs. These techniques provided direct elemental confirmation of the metal content in the samples, which is especially crucial in complex plant matrices where optical signals can be masked by flavonoids, proteins, or residual plant matter. EDS offers localised elemental identification at the microscale during transmission electron microscopy, while LIBS allows for rapid elemental profiling of the bulk sample. Together, they address the limitations of indirect evidence from optical techniques and enhance the reliability of NP identification. The EDS and LIBS findings confirmed that both *M. sativa* genotypes could mediate the synthesis of AgNPs and AuNPs via green chemistry mechanisms. Notably, the high relative weight percentages of Ag and Au in the AgNO_3_- and HAuCl_4_-treated samples, respectively, underscore the efficiency of the reduction process. Moreover, the absence of these metals in the control samples excludes contamination and supports the specificity of ion incorporation in the synthesised nanostructures. The presence of K, Ca, and P, although low, may be related to the residual ionic content of the culture medium or natural components of the plant tissue.

These findings further validate that nanoparticle formation occurred through biological reduction rather than physical contamination or precursor residue interference.

X-ray diffraction (XRD) analysis was performed to investigate the crystalline structure of the synthesised nanoparticles. Although the analysis confirmed the presence of solid-phase nanostructures, the diffraction patterns were characterised by very weak and broadened reflections, with intensities close to the instrumental noise level. This limited resolution is likely attributed to the small size of the particles and partial structural disorder. It is well-documented that nanocrystals below ~20–30 nm often produce weak, broadened peaks that challenge the detection limits of conventional laboratory XRD instruments [82,83]. The observed diffraction features—most notably the broadened (111) reflection in Ag samples and weak signals from Au nanoparticles—are therefore consistent with the expected nanoscale dimensions. Complementary transmission electron microscopy (TEM) analysis confirmed the presence of quasi-spherical nanoparticles in the 5–25 nm range, validating successful nanoparticle synthesis despite XRD-related limitations.

This study contributes significantly to the field of nanobiotechnology by presenting a green, safe, and cost-effective method for synthesising AgNPs and AuNPs. Its novelty lies in the use of callus extracts from two distinct *M. sativa* genotypes, ‘Kometa’ and ‘La Bella Campagnola’, to biosynthesise spherical AgNPs and AuNPs. The comparative approach revealed genotype-dependent differences in nanoparticle yield and stability, highlighting the role of biochemical composition in green synthesis efficiency. The main scientific contribution of this work is the development of an optimised biosynthesis protocol using plant callus cultures as a controlled and reproducible biological system [16,17]. For future studies, additional sample purification, such as dialysis, should be employed before transmission electron microscopy to reduce particle overlap and to improve measurement accuracy. It is also important to isolate the synthesised NPs from the extract for downstream applications. Further investigation is needed to evaluate the potential biological activity of the synthesised nanoparticles, including antibacterial effects, in future studies and to compare their bioactivity with that of chemically synthesised NPs [16,17,18,83]. In contrast to conventional leaf or seedling extracts, callus cultures provide a stable and standardised metabolite source, reducing biochemical variability and improving synthesis reproducibility.

## 4. Materials and Methods

### 4.1. Medicago sativa Seedling Cultivation

Seeds of a Spanish genotype (‘La Bella Campagnola’ certified seeds C/2 B25658202200005) and Polish genotype (‘Kometa’) of *M. sativa* were obtained from DANKO Plant Breeding Sp. z o.o. (Choryń 27, 64-000 Kościan, Poland). The seeds were rinsed with deionised water and then soaked in 3% sodium hypochlorite for 20 min. They were rinsed at least five times with sterilised, autoclaved deionised water (120 °C, 1 atm) [25,50]. Germination took place in water on filter paper to eliminate the influence of external nutrients on seedling growth. After approximately 14 days of germination and growth, the seedlings were transferred to a growth chamber at 24 °C and cultivated in standard Murashige and Skoog (MS) medium [48]. A total of 15 seedlings were grown for each genotype.

### 4.2. Medicago sativa Callus Culture Cultivation

To obtain callus cultures for both cultivars, explants from 30 seedlings (1 seedling per callus culture) were cultivated on basal MS medium supplemented with 3% sucrose, 0.7% agar (pH 5.8), 1 mg/L of 2,4-dichlorophenoxyacetic acid (2,4-D), and 1 mg/L of 6-benzylaminopurine (BAP) [48]. The cultures were maintained for 4 weeks.

### 4.3. Preparation of M. sativa Callus Culture Extract for AgNP and AuNP Biosynthesis

Callus cultures were cut into small pieces, and 2 g of callus tissue was weighed for each group. Then, 100 mL of deionised water was added to the weighed callus tissue. The mixture was ground with a pestle and heated at 60 °C for 3 h, followed by cooling [20]. For NP biosynthesis, the obtained extracts were divided into three groups: a control group without any precursor, and experimental groups treated with silver nitrate (AgNO_3_) or chloroauric acid (HAuCl_4_) for 24 or 48 h. Then, 1000 mg/L AgNO_3_ stock solution was prepared by dissolving 0.02 g of AgNO_3_ powder in 20 mL of deionised water, and a 200 mg/L HAuCl_4_ stock solution was prepared by dissolving 0.004 g of HAuCl_4_ powder in 20 mL of deionised water. For NP synthesis, 10 mL of each stock solution was added to 100 mL of callus extract, and the mixtures were manually agitated for 10 min to ensure homogeneous mixing of the precursors. The pH of all mixtures was adjusted to 7.0 prior to incubation. The resulting mixtures were filtered using a new filter for each sample and incubated in the dark for 30 min before centrifugation [31].

The precursor concentrations applied in this study were selected based on previously reported green synthesis protocols. A concentration of 1000 mg/L AgNO_3_ (≈5.9 mM) falls within the commonly used range of 0.5–5 mM, which has been shown to efficiently promote silver nanoparticle formation in plant and callus extract systems [24,27,40,49]. Similarly, 200 mg/L HAuCl_4_ (≈0.6 mM) was chosen based on earlier studies where gold nanoparticle biosynthesis was successfully achieved using 0.5–1 mM HAuCl_4_ [18,33,45]. These concentrations ensured effective nanoparticle formation without causing precursor-induced precipitation or instability.

### 4.4. Experimental Replication

All measurements were performed using independent samples, and each sample was analysed in triplicate (three analytical repetitions) to ensure reproducibility and minimise measurement error.

### 4.5. Determination of Light Absorbance Intensity

A NanoDrop 1000 spectrophotometer (Thermo Fisher Scientific, Wilmington, NC, USA) was used to measure the light absorbance of the suspension solution. Each sample was analysed in triplicate. The ND-1000 V3.6.0 software was used to process the absorbance data obtained at wavelengths of 350 and 470 nm for the AgNO_3_-treated samples and 450 and 650 nm for the HAuCl_4_-treated samples [50,51].

### 4.6. Determination of Light Absorbance

A two-beam UV-2550PC spectrophotometer (Shimadzu, Kyoto, Japan) was used to measure the light absorbance across a wavelength range from 190 to 900 nm. The ultraviolet–visible (UV–Vis) spectrum indicated AgNP formation at a wavelength of 286 nm and AuNP formation at 285 and 534 nm [20,52]. Absorbance peaks were recorded when intense absorption occurred, and light absorbance measurements were taken for samples of both genotypes.

### 4.7. Transmission Electron Microscopy Visualisation of M. sativa Callus Culture Extract

Alfalfa callus culture extracts were visualised using a transmission electron microscope (JEM-1220, JEOL, Tokyo, Japan) operating at an accelerating voltage of 80 kV. Samples for TEM observations were prepared by placing droplets of the hydrocolloid suspensions onto Formvar-coated 3 mm 200-Mesh Cu grids (Agar Scientific, Stansted, UK). Immediately after air drying, the grids were examined without staining. Images were captured with an exposure time of 1.2 s, using a Morada CCD digital camera (Olympus SIS, Münster, Germany) [53,54,55,56]. The control, AgNO_3_-treated, and HAuCl_4_-treated sample extracts were prepared by depositing hydrocolloid droplets onto Formvar-coated copper grids (Agar Scientific Ltd., Stansted, UK). Once the droplets had air-dried, the grids were inserted into the transmission electron microscope for imaging.

### 4.8. Dynamic Light Scattering (DLS) and Zeta Potential

DLS was employed to determine the particle size distribution. The surface charge of the obtained NPs was assessed by measuring the zeta potential using electrokinetic techniques. These measurements aimed to evaluate the stability of the surface charge in the hydrosols. The analyses were conducted using a Zetasizer Nano series (Malvern Panalytical, Malvern, UK) at 25 °C. For each sample, the average hydrodynamic diameter (Z-average) and polydispersity index (PDI) were measured three times, while the zeta potential was measured four times.

### 4.9. NP Size Distribution Analysis by Nanoparticle Tracking Analysis (NTA)

The hydrodynamic diameter and concentration of AgNPs and AuNPs were determined using NTA performed with the NanoSight Pro system (Malvern Panalytical). This technique enables direct visualisation and tracking of NPs in suspension based on their Brownian motion, thereby allowing precise size distribution and concentration measurements.

#### 4.9.1. Sample Preparation

NPs were suspended in ultrapure Milli-Q water and vortexed gently to ensure uniform dispersion. Prior to analysis, samples were filtered through a 0.22 μm polyvinylidene difluoride syringe filter to remove dust and aggregates. To ensure optimal particle count per frame (i.e., 20–120 particles), each sample was diluted appropriately (typically 1:100 to 1:10,000, depending on the stock concentration) using ultrapure water. All measurements were performed at 25 ± 1 °C.

#### 4.9.2. Instrument Settings and Measurement Protocol

For each sample, 1 mL of diluted NP suspension was loaded into the instrument’s flow cell using a sterile syringe. The syringe pump was used to maintain a constant flow during the measurement, and a 1 s flow was applied prior to each video capture to ensure even particle distribution and to minimise particle settling or aggregation. The 488 nm laser was used for both AuNPs and AgNPs. The camera level was adjusted automatically by the NanoSight software (version 3.4) to optimise contrast while avoiding oversaturation.

### 4.10. Elemental Analysis by Energy-Dispersive X-Ray Spectroscopy (EDS)

EDS was performed using Scanning Electron Microscopy (SEM) (Quanta 250 FEG SEM, FEI, Hillsboro, OR, USA) detector coupled with a Quanta 250 FEG scanning electron microscope (Quanta 250 FEG SEM, FEI, Hillsboro, OR, USA), an equipped with Energy Dispersive X-ray spectroscopy detector (EDS-EDAX, LLC, Mahwah, NJ, USA) to confirm the elemental composition of biosynthesised NPs. The analysis was conducted separately for AgNPs and AuNPs synthesised from *M. sativa* ‘Kometa’ and ‘La Bella Campagnola’ callus extracts. After centrifugation and removal of unreacted precursors, NP pellets were deposited on carbon-coated scanning electron microscopy stubs and dried under sterile laminar flow for 24 h. Spectra were acquired under high vacuum at a 20 kV accelerating voltage, with 20 random spots analysed per sample. Elemental distribution maps and quantitative profiles were obtained using the EDAX Genesis software ver. 5.2. This method allowed precise verification of Ag and Au incorporation into NPs, confirming successful biosynthesis in both *M. sativa* genotypes.

### 4.11. Laser-Induced Breakdown Spectroscopy (LIBS) Analysis

LIBS was performed to further verify the elemental composition and to detect trace levels of precursor metals and matrix elements. NP suspensions derived from the *M. sativa* ‘Kometa’ and ‘La Bella Campagnola’ callus extracts were independently spotted on silicon wafers (2-inch Dummy CZ-Si, Microchemicals, Ulm, Germany) and vacuum-dried at 50 °C for 24 h. The LIBS system employed a Nd:YAG laser (Quantel Brio, Lannion, France) at 1064 nm, a 4 ns pulse width, 46 mJ energy per pulse, and a gate delay of 500 ns. Plasma emission was collected via optical fibre and analysed with a broadband spectrometer (Avantes, Apeldoorn, The Netherlands).

### 4.12. X-Ray Diffraction (XRD) Analysis

X-ray diffraction (XRD) analysis was performed to investigate the crystalline structure and phase composition of the synthesised nanoparticles. The presence and crystallographic structure of Ag and Au nanoparticles were investigated by means of X-ray studies. Diffraction measurements for samples obtained from mentioned suspensions were carried out using the BRUKER D8 Discover diffractometer (Bruker AXS GmbH, Karlsruhe, Germany) equipped with a CuK_α_ radiator (λK_α1_ = 1.54056 Å, λK_α2_ = 1.54443 Å, Siemens KFL CU 2 K, 40 kV voltage, and 40 mA current in operating mode), Göbel FGM2 mirror, and 1D LYNXEYE detector (Bruker AXS GmbH, Karlsruhe, Germany). The Bragg–Brentano diffraction geometry was applied. The diffraction angle 2θ was scanned with a step of 0.0152° and acquisition time of 1 s per step. The studies were carried out at RT in the temperature-stabilised Anton Paar HTK 1200N chamber (Anton Paar GmbH, Graz, Austria). Phase analysis was performed using the Match! ver. 4.2 application, provided by Crystal Impact, with support of the COD database.

### 4.13. Statistical Analysis

Each treatment was performed in triplicate to ensure data reliability. The data were analysed using Microsoft Excel (Microsoft Office Professional Plus 2019) and are presented as the mean ± standard deviation (SD). For each genotype, a paired two-sample *t*-test was performed to determine significant differences in light absorbance between control samples and those treated with AgNO_3_ or HAuCl_4_ at specific wavelengths. In addition, a one-way analysis of variance (ANOVA) was applied to compare differences between exposure times (24 h and 48 h). For all statistical analyses, differences were considered statistically significant at *p* < 0.05.

## 5. Conclusions

In this study, callus cultures were successfully obtained from 2-week-old *M. sativa* ‘Kometa’ and ‘La Bella Campagnola’ seedlings, which effectively absorbed precursors (AgNO_3_ and HAuCl_4_) into their cells. The sample preparation and visualisation methods were successfully adapted and improved, revealing the presence of both AgNPs and AuNPs in the callus culture extracts after biosynthesis. To improve image quality and obtain more accurate data, transmission electron microscopy (TEM) analysis was repeated, confirming the presence of quasi-spherical nanoparticles with diameters in the range of 5–25 nm for both AuNPs and AgNPs, depending on the genotype. NPs were detected using transmission electron microscopy, which revealed changes in the zeta potential and particle dispersion in the solution, indicating precursor breakdown. Both large NP agglomerates and small nanoparticles were obtained and visualised.

Spectrophotometric analysis confirmed significant (*p* < 0.05) changes in light absorbance of the samples exposed to the metal precursors compared with the control samples. Based on the spectral data, HAuCl_4_ was more efficient in the formation of new particles, whereas AgNO_3_ remained primarily in an ionic or small-cluster form rather than forming stable, well-defined plasmonic NPs. Although a typical SPR peak for AgNPs was not clearly observed, the combined transmission electron microscopy results, zeta potential shifts, and NTA results strongly support the successful biosynthesis of AgNPs and AuNPs, particularly in the ‘Kometa’ callus culture extract. EDS and LIBS provided the most conclusive evidence of NP biosynthesis. These techniques clearly confirmed the presence of Ag and Au only in the samples treated with metal precursors, and not in the control samples. This rules out contamination. Taken together with UV–Vis spectra, solution colour changes, and control NTA experiments, these results provide a robust, multi-method confirmation of nanoparticle formation.

However, direct crystallographic characterisation by X-ray diffraction (XRD) analysis revealed only weak and significantly broadened reflections, with signal intensities approaching the detection limit of the instrument. These features are indicative of nanoscale crystallites with reduced coherence lengths and pronounced surface disorder, which are known to cause substantial peak broadening and intensity suppression in XRD patterns of particles below ~30 nm. The absence of sharp, well-defined diffraction peaks is therefore consistent with the particle size distribution confirmed by repeated TEM imaging, which showed spherical nanoparticles in the range of 5–25 nm. Moreover, the presence of broadened reflections—particularly from the (111) lattice plane in Ag samples—suggests partial crystallinity, despite the limitations of laboratory-scale XRD resolution at the nanoscale. Through XRD was limited by particle size, the combination of complementary methods—both qualitative (colour change, UV–Vis) and quantitative/imaging (repeated TEM with improved resolution, DLS, NTA, EDS, LIBS)—provides a strong set of indirect and direct evidence for the formation and presence of AuNPs and AgNPs in this biosynthesis system.

This study contributes to nanobiotechnology by developing an environmentally sustainable and cost-effective method for synthesising quasi-spherical AgNPs and AuNPs using *M. sativa* callus culture extracts. It also provides valuable insights into genotype-specific variations in NP biosynthesis. The repeated TEM analysis with improved image quality confirmed the synthesis of very small nanoparticles (5–25 nm), which explains the absence of detectable XRD signals and validates the successful formation of nanostructures despite instrumental limitations. However, a methodological limitation of this study is the absence of additional sample purification prior to analysis, which may have contributed to nanoparticle aggregation and partial particle overlap during imaging. To improve nanoparticle separation and enable more detailed structural characterisation, future research should incorporate purification steps such as dialysis or density gradient centrifugation. Overall, these findings strengthen the conclusion that M. sativa callus cultures can serve as effective biotemplates for nanoparticle production and may enable the development of plant-based nanomaterials for biomedical or environmental applications. 

## Figures and Tables

**Figure 1 ijms-26-10772-f001:**
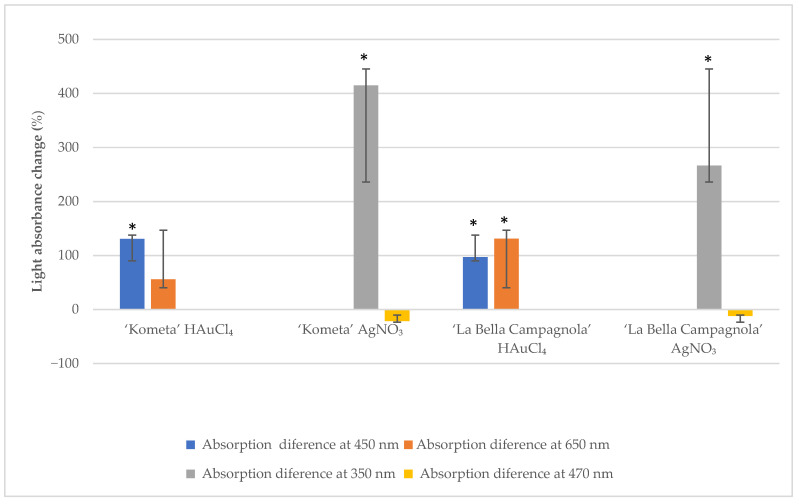
Light absorbance change (relative to the control) at different wavelengths after exposure to the precursor for 24 h. An asterisk indicates a significant difference compared with the control (*p* < 0.05, paired two-sample *t*-test).

**Figure 2 ijms-26-10772-f002:**
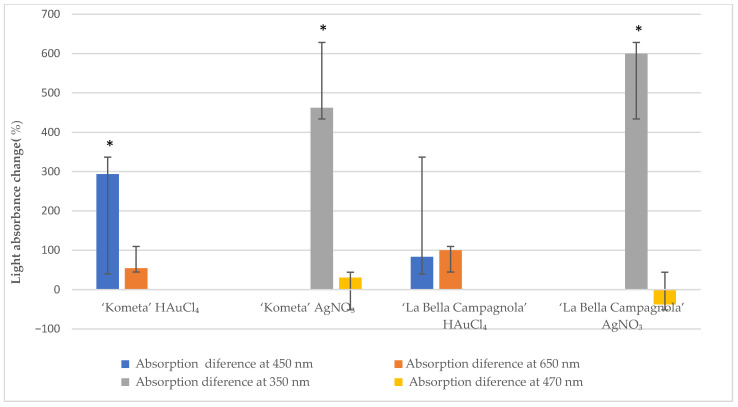
Light absorbance change (relative to the control) at different wavelengths after exposure to the precursor for 48 h. An asterisk indicates a significant difference compared with the control (*p* < 0.05, paired two-sample *t*-test).

**Figure 3 ijms-26-10772-f003:**
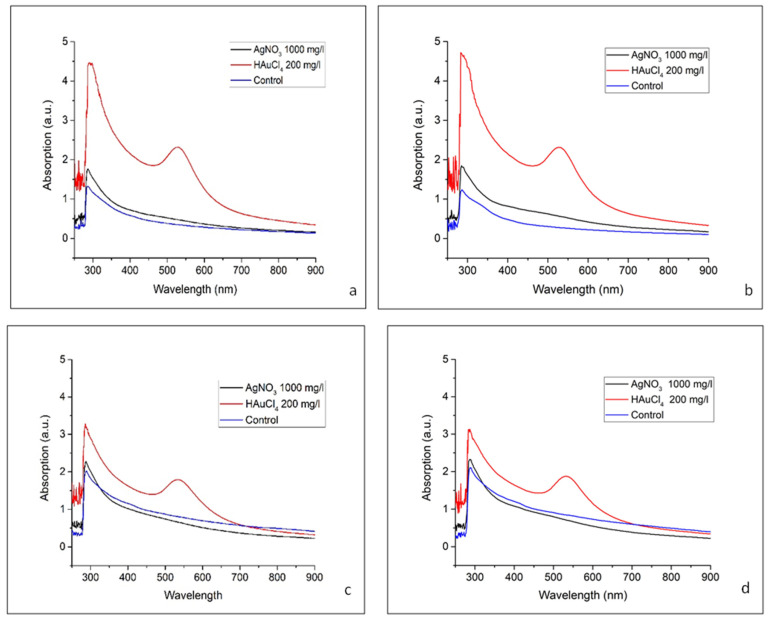
Absorbance at different peaks for *Medicago sativa* ‘Kometa’ and ‘La Bella Campagnola’ samples exposed to AgNO_3_ and HAuCl_4_ for 24 h (**a**,**c**) and 48 h (**b**,**d**), respectively.

**Figure 4 ijms-26-10772-f004:**
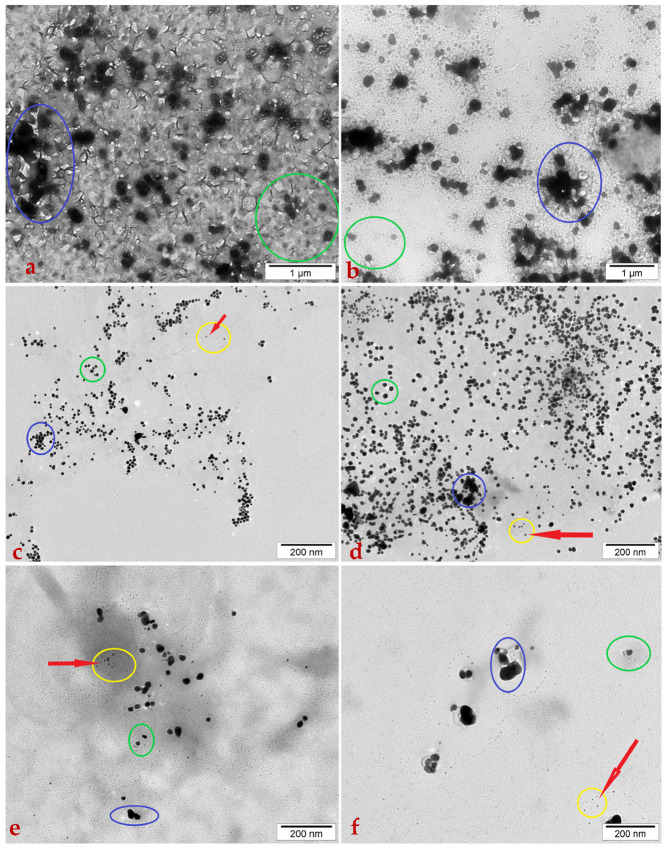
Transmission electron micrographs of the *Medicago sativa* ‘Kometa’ control sample (**a**); the *M. sativa* ‘La Bella Campagnola’ control sample (**b**); the ‘Kometa’ callus culture extract treated with 200 mg/L HAuCl_4_ (**c**); the ‘La Bella Campagnola’ callus culture extract treated with 200 mg/L HAuCl_4_ (**d**); the ‘Kometa’ callus culture extract treated with 1000 mg/L AgNO_3_ (**e**); and the ‘La Bella Campagnola’ callus culture extract treated with 1000 mg/L AgNO_3_ (**f**). The blue, green, and yellow lines indicate structures of different sizes in the sample. The smaller black dots, outlined by a yellow line, are considered nanoparticles. The red arrows point to nanoparticles.

**Figure 5 ijms-26-10772-f005:**
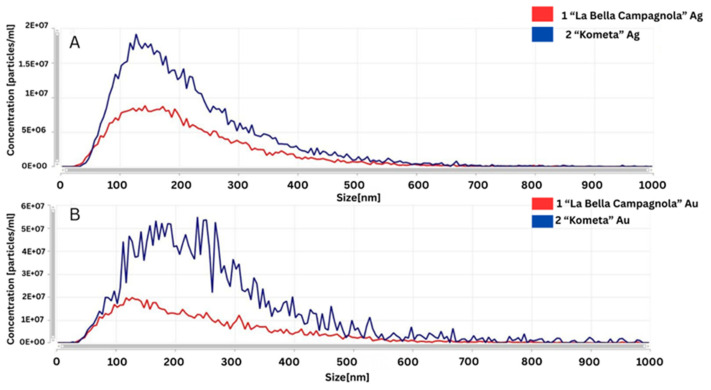
Comparative Nanoparticle Tracking Analysis of silver (Ag) (**A**) and gold (Au) (**B**) nanoparticles synthesised in the ‘*La Bella Campagnola*’ and ‘*Kometa*’ callus culture extracts.

**Figure 6 ijms-26-10772-f006:**
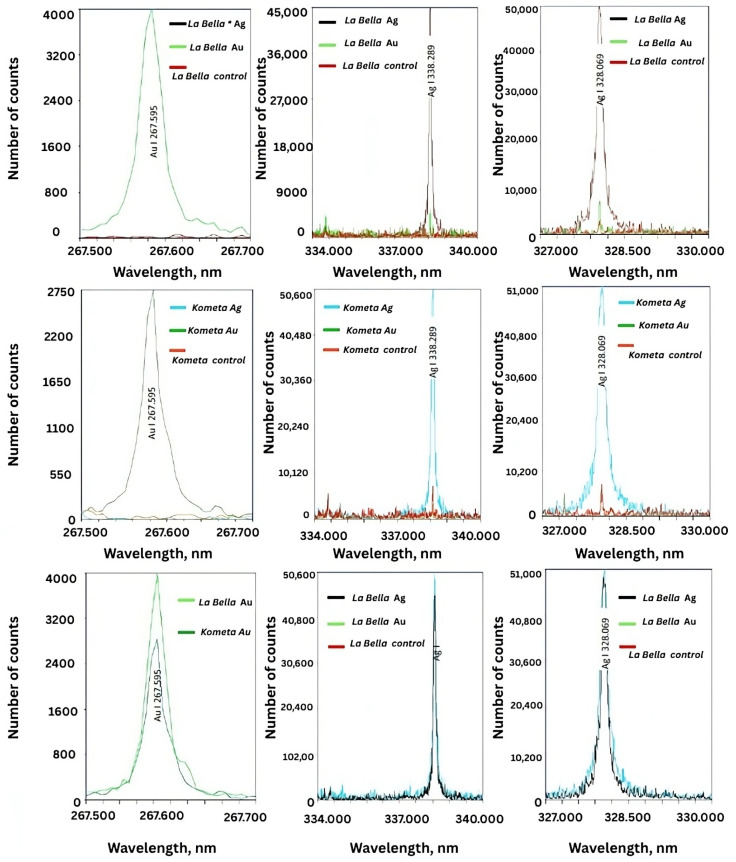
Representative laser-induced breakdown spectra of biogenic silver (Ag) and gold (Au) nanoparticles in *Medicago sativa* ‘Kometa’ and ‘La Bella Campagnola’ callus culture extracts. Note that ‘La Bella Campagnola’ is shortened to La Bella on the graphs.

**Figure 7 ijms-26-10772-f007:**
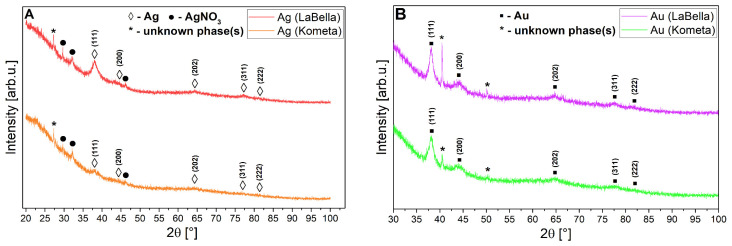
A set of XRD patterns for Ag samples (**A**) and Au (**B**) prepared from ‘LaBella Campagnola’ and ‘Kometa’ sources.

**Table 1 ijms-26-10772-t001:** Absorbance at different peaks for *Medicago sativa* ‘Kometa’ and ‘La Bella Campagnola’ samples after exposure to AgNO_3_ or HAuCl_4_ for 24 or 48 h.

Precursor and Exposure Time (h)	Precursor Concentration (mg/L)	Absorbance (a.u.) at 286 nm for ‘La Bella Campagnola’	Absorbance (a.u.) Peak at 534 nm for ‘La Bella Campagnola’	Absorbance (a.u.) at 286 nm for ‘Kometa’	Absorbance (a.u.) at 534 nm for ‘Kometa’
24 h					
Control		2.0	-	1.3	-
HAuCl_4_	200	3.3	1.8	4.4	2.3
AgNO_3_	1000	2.3	-	1.8	-
48 h					
Control		2.1	-	1.2	-
HAuCl_4_	200	3.1	1.9	4.7	2.3
AgNO_3_	1000	2.3	-	1.8	-

**Table 2 ijms-26-10772-t002:** Summary of the physicochemical characterisation data for the biosynthesised silver nanoparticles (AgNPs) and gold nanoparticles (AuNPs) in Milli-Q water.

Sample	NPs	Z-Average from DLS	Polydispersity Index	Zeta Potential (mV)	Size Based on Microscopy (nm)
‘Kometa’ Control	-	260 ± 70	0.28 ± 0.03	−10. ± 3	-
‘Kometa’ HAuCl_4_	Au	1074 ± 10 ↑	0.61 ± 0.05 ↑ *	−16 ± 1 ↓ *	28 ± 18
‘Kometa’ AgNO_3_	Ag	324 ± 53 ↑	0.44 ± 0.03 ↑ *	−15 ± 1 ↓ *	27 ± 4
‘La Bella Campagnola’ Control	-	551 ± 20	0.33 ± 0.05	−13 ± 1	-
‘La Bella Campagnola’ HAuCl_4_	Au	1074 ± 19 ↑	0.54 ± 0.03 ↑ *	−17± 1 ↓ *	17 ± 7
‘La Bella Campagnola’ AgNO_3_	Ag	320 ± 9 ↓	0.44 ± 0.07 ↑ *	−16 ± 1 ↓ *	58 ± 8

The data are presented as mean ± standard deviation. ↑ and ↓ indicate an increase or decrease, respectively, compared with the control. * A significant difference compared with the control (*p* < 0.05, paired two-sample *t*-test).

**Table 3 ijms-26-10772-t003:** Summary of the size distribution data obtained from Nanoparticle Tracking Analysis.

Sample	Mode (nm)	Mean (nm) ± 95% CI (From–To)	D90 (nm)	Concentration (Particles/mL)
‘Kometa’ AgNO_3_	128 ± N/A	220 ± (218–222)	385 ± N/A	7 × 10^8^ ± N/A
‘Kometa’ HAuCl_4_	128 ± N/A	243 ± (240–246)	440 ± N/A	9 × 10^8^ ± N/A
‘Kometa’ Control	253 ± N/A	284 ± (281–287)	492 ± N/A	6 × 10^8^ ± N/A
‘La Bella Campagnola’ AgNO_3_	143 ± N/A	219 ± (217–221)	384 ± N/A	4 × 10^8^ ± N/A
‘La Bella Campagnola’ HAuCl_4_	173 ± N/A	242 ± (239–245)	449 ± N/A	4 × 10^8^ ± N/A
‘La Bella Campagnola ‘Control	183 ± N/A	238 ± (235–241)	404 ± N/A	6 × 10^8^ ± N/A

Values represent mode, D90, and mean (95% confidence interval, from–to). The 95% CI was calculated using the normal distribution (z = 1.96).

**Table 4 ijms-26-10772-t004:** Elemental composition (wt%) of the ‘Kometa’ and ‘La Bella Campagnola’ samples based on energy-dispersive X-ray spectroscopy.

Elemental Weight [%]	‘Kometa’ Control	‘Kometa’ HAuCl_4_	‘Kometa’ AgNO_3_	‘La Bella Campagnola’ Control	‘La Bella Campagnola’ HAuCl_4_	‘La Bella Campagnola’ AgNO_3_
C	40.5 ± 1	43.1 ± 1	40.9 ± 0.9	40.6 ± 1	42.8 ± 1	39.2 ± 0.8
O	24 ± 1	2.1 ± 0.2	1.9 ± 0.1	23.25 ± 1	2.1 ± 0.1	1.8 ± 0.1
Na	1.8 ± 0.2	0.85 ± 0.07	0.93 ± 0.06	1.5 ± 0.2	0.81 ± 0.05	0.88 ± 0.06
Mg	0.18 ± 0.05	0.14 ± 0.03	0.17 ± 0.04	0.22 ± 0.04	0.15 ± 0.03	0.19 ± 0.04
Si	27.5 ± 2	3.6 ± 0.4	4 ± 0.5	28 ± 1	3 ± 0.3	4.1 ± 0.6
Cl	0.3 ± 0.1	9.9 ± 0.8	5.7 ± 0.6	0.5± 0.1	10.2 ± 0.9	5.3 ± 0.7
K	2.5 ± 0.3	11.9 ± 1.2	7.4 ± 1.1	2.1 ± 0.4	12 ± 1	7.9 ± 1
Ca	1.11 ± 0.2	2.4 ± 0.3	1.9 ± 0.2	1.1 ± 0.3	2.4 ± 0.2	2.1 ± 0.3
Au	0.00	25 ± 3	0.00	0.00	26 ± 2	0.00
Ag	0.00	0.00	39 ± 3	0.00	0.00	41 ± 3

The data are presented as the mean ± standard deviation.

## Data Availability

The data presented in this study are available on request from the corresponding author.

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
