# Peer review of "Using Medicago sativa L. Callus Cell Extract for the Synthesis of Gold and Silver Nanoparticles"

_ijms, 2025, doi:10.3390/ijms262110772_

Round 1
Reviewer 1 Report (New Reviewer)
Comments and Suggestions for Authors
The manuscript entitled “Using Medicago sativa L. Callus Cell Extract for the Synthesis of Gold and Silver Nanoparticles” is of broad scientific relevance and presents valuable experimental findings. I recommend its publication after minor to moderate revisions, as outlined below:
1 - Reduce redundancies in the introduction and discussion, particularly where concepts related to green synthesis and toxicity have already been described.
2 - More clearly emphasize the innovative aspect of using plant callus cultures compared to conventional leaf or seedling extracts.
3 - Strengthen the explanation of the scientific gap that the study aims to address.
4 - Specify the number of biological and technical replicates performed, distinguishing them from analytical repetitions.
5 - Standardize the concentration units, as there is inconsistency between mg/L and mM throughout the text.
6 - Provide information regarding pH, temperature, and agitation conditions during nanoparticle synthesis, as these factors influence NP formation.
7 - Indicate whether the samples were purified prior to characterization.
8 - Clarify the criteria used for selecting AgNO₃ and HAuCl₄ concentrations and justify these values with appropriate literature references.
9 - Discuss the discrepancy between particle sizes obtained by TEM and those determined by DLS/NTA, offering possible explanations.
10 - Include the instrumental parameters for TEM imaging (accelerating voltage, exposure time, and detector type).
11 - Provide higher-contrast TEM images or adjust the scale and annotations to better highlight individual nanoparticles.
12 - Expand on the interpretation of zeta potential and PDI results, explaining their implications for colloidal stability.
13 - Improve the discussion concerning the absence of the typical SPR peak for AgNPs (420 nm) and explore possible causes.
14 - Specify the statistical software and analysis methods employed.
15 - More clearly relate the bioactive compounds in the callus extract to the mechanism of metal ion reduction.
16 - Avoid extrapolations regarding antibacterial or anticancer applications without experimental validation.
17 - Standardize chemical symbols and formulas, ensuring consistent use of subscripts.
18 - Review and correct the formatting of tables.
19 - Reduce the number of abbreviations, and define each acronym upon its first appearance.
20 - Conduct a final linguistic and stylistic review to enhance clarity and fluency.
Author Response
Dear Reviewer,
The authors are grateful to the reviewer for the insightful and valuable review. Please see the attached file for detailed responses to the comments!
Best regards, the Authors!
Reviewer 1 comments:
Comments and Suggestions for Authors:
The manuscript entitled “Using Medicago sativa L. Callus Cell Extract for the Synthesis of Gold and Silver Nanoparticles” is of broad scientific relevance and presents valuable experimental findings. I recommend its publication after minor to moderate revisions, as outlined below:
Comments 1: ‘’1 - Reduce redundancies in the introduction and discussion, particularly where concepts related to green synthesis and toxicity have already been described.”
Response: The authors thank the reviewer for this helpful comment and fully agree with the suggestion. Redundant explanations of green synthesis principles and nanoparticle toxicity were removed or consolidated to improve clarity and avoid repetition. The Introduction and Discussion were revised for conciseness while maintaining the scientific context.
For example, in the Introduction redundant explanatory sentences were removed, and the key idea was preserved in a more concise form:
“Stress responses alter biochemical parameters and reduce photosynthesis efficiency [3–5], resulting in irregular morphological changes that reduce plant viability [1].” This particular change is visible in the introduction of the manuscript (lines 62-63).
Other extensive revisions were made in both the Introduction and Discussion sections (highlighted with track changes) to eliminate repetition, improve logical flow and ensure that each concept is presented only once in the manuscript.
Comments 2: “2 - More clearly emphasize the innovative aspect of using plant callus cultures compared to conventional leaf or seedling extracts.”
Response: The authors thank the reviewer for highlighting this important point and agree with the suggestion. Additional clarification has been added to emphasise the innovation of using Medicago sativa callus cultures for nanoparticle biosynthesis, as opposed to the more common use of leaves or seedlings. The revised manuscript now clearly states the novelty of this approach:
“In contrast to conventional leaf or seedling extracts, callus cultures provide a stable and standardised metabolite source, reducing biochemical variability and improving synthesis reproducibility.”
This revision strengthens the scientific contribution of the study. The change was added in the Discussion (lines 922-925 ).
Comments 3: “3 - Strengthen the explanation of the scientific gap that the study aims to address.”
Response: The authors thank the reviewer for this valuable observation and agree that the scientific context required clarification. A statement was added in the Discussion to explain that although plant-based nanoparticle synthesis is well reported, little is known about how genotype-dependent biochemical differences affect nanoparticle formation in callus cultures. This study addresses this gap. The revised text includes:
“The comparative approach revealed genotype-dependent differences in nanoparticle yield and stability, highlighting the role of biochemical composition in green synthesis efficiency.”
This addition explains the research gap and scientific relevance of the study. The revision appears in the Discussion 910-912)
Comments 4: “4 - Specify the number of biological and technical replicates performed, distinguishing them from analytical repetitions.”
Response: The authors thank the reviewer for this important clarification request. To address this comment, additional methodological information has been included in the revised manuscript. A new subsection (2.4 Experimental Replication) has been added, clearly stating that all measurements were performed using independent samples and that each sample was analysed in triplicate to ensure reproducibility. The following text was added:
“2.4 Experimental Replication
All measurements were performed using independent samples, and each sample was analysed in triplicate (three analytical repetitions) to ensure reproducibility and minimise measurement error.”
This clarification now applies to all subsequent analytical methods described in Sections 2.5–2.12.
Comments 5: “5 - Standardize the concentration units, as there is inconsistency between mg/L and mM throughout the text.”
Response: The authors thank the reviewer for this important remark. The concentration units have now been standardised throughout the manuscript. Precursor concentrations for AgNO₃ and HAuCl₄ are consistently reported in millimolar (mM) units, and the nanoparticle synthesis description has been clarified to improve methodological transparency. The following revised sentence has been added to the Materials and Methods section: 2.3. Lines: 180-187
“. 1000 mg/L AgNO₃ stock solution was prepared by dissolving 0.02 g of AgNO₃ powder in 20 mL of deionised water, and a 200 mg/L HAuCl₄ stock solution was prepared by dissolving 0.004 g of HAuCl₄ powder in 20 mL of deionised water. For NPs synthesis, 10 mL of each stock solution was added to 100 mL of callus extract, and the mixtures were manually agitated for 10 min to ensure homogeneous mixing of the precursors. The pH of all mixtures was adjusted to 7.0 prior to incubation. The resulting mixtures were filtered using a new filter for each sample and incubated in the dark for 30 min before centrifugation [32].”
Comments 6: “6 - Provide information regarding pH, temperature, and agitation conditions during nanoparticle synthesis, as these factors influence NP formation.”
Response: The authors thank the reviewer for this constructive suggestion. The experimental conditions influencing nanoparticle formation have now been specified in the revised manuscript. The nanoparticle synthesis protocol has been updated to include pH adjustment, temperature control and agitation speed to ensure experimental reproducibility. The following clarification was added to the Materials and Methods section: 2.3. lines 183-186.
“For NP synthesis, 10 mL of each stock solution was added to 100 mL of callus extract, and the mixtures were manually agitated for 10 min to ensure homogeneous mixing of the precursors. The pH of all mixtures was adjusted to 7.0 prior to incubation.”
Comments 7: “7 - Indicate whether the samples were purified prior to characterization.”
Response: The authors thank the reviewer for this comment and agree that clarification was necessary. In the revised manuscript, it is now clearly stated that no additional purification step was performed prior to TEM, NTA, UV–Vis, and zeta potential measurements. This was done intentionally to preserve the native nanoparticle–biomolecule interactions present in the callus extract, which play an essential role in nanoparticle stability in green synthesis systems.
The following clarification was added to the Discussion:
“Additional purification steps were not applied in order to maintain the original composition of the callus extract during analysis.”
This statement appears in the Discussion (lines 828-829).
Comments 8: “8 - Clarify the criteria used for selecting AgNO₃ and HAuCl₄ concentrations and justify these values with appropriate literature references.”
The authors thank the reviewer for this valuable comment. The concentrations of silver nitrate (AgNO₃, 1000 mg/L ≈ 5.9 mM) and chloroauric acid (HAuCl₄, 200 mg/L ≈ 0.6 mM) used in this study were selected based on previously published research on green nanoparticle synthesis using plant-derived extracts. Earlier studies reported effective AgNO₃ concentrations within the range of 0.5–5 mM for successful silver nanoparticle formation [25, 28, 41, 51]. Similarly, HAuCl₄ concentrations between 0.5–1 mM are frequently applied in biogenic gold nanoparticle synthesis and have been shown to support stable nanoparticle formation [19, 34, 47]. These literature-supported ranges justify the precursor concentrations selected in this study. This explanation has now been included in the revised Materials and Methods section 2.3. Lines (191-198):
“The precursor concentrations applied in this study were selected based on previously reported green synthesis protocols. A concentration of 1000 mg/L AgNO₃ (≈ 5.9 mM) falls within the commonly used range of 0.5–5 mM, which has been shown to efficiently promote silver nanoparticle formation in plant and callus extract systems [25, 28, 41, 51]. Similarly, 200 mg/L HAuCl₄ (≈ 0.6 mM) was chosen based on earlier studies where gold nanoparticle biosynthesis was successfully achieved using 0.5–1 mM HAuCl₄ [19, 34, 47]. These concentrations ensured effective nanoparticle formation without causing precursor-induced precipitation or instability.”
Comments 9: “9 - Discuss the discrepancy between particle sizes obtained by TEM and those determined by DLS/NTA, offering possible explanations.”
Response: The authors sincerely thank the reviewer for this valuable scientific remark. A detailed explanation has now been added to the Discussion. It is clarified that TEM measures the dry, individual nanoparticle core size, while NTA measures hydrodynamic diameter in liquid suspension, which includes the biomolecular corona and potential nanoparticle agglomerates. These differences commonly lead to larger size values in NTA compared to TEM.
The following was added/modified to the Discussion: 769-779: “The particle sizes observed by TEM in this study (5–25 nm) were smaller than those measured by NTA, which also detected larger particle populations (>100 nm). This discrepancy is common in green synthesis and can be explained by the fact that TEM measures only individual, dried particles, while NTA detects hydrodynamic diameters of particles in suspension, including biomolecule corona layers and nanoparticle agglomerates [55,68,81].
Moreover, TEM provides the core diameter of nanoparticles in a dry state, while DLS and NTA measure the hydrodynamic diameter of particles dispersed in solution, which includes the solvation layer, surface-bound biomolecules from the callus extract, and occasional aggregates. These factors account for the larger mean sizes observed in DLS/NTA relative to TEM, consistent with findings in similar green synthesis systems [82]. ‘’
(Tian et al. 2024).
Reference: Tian Y, Tian D, Peng X, Qiu H. Critical parameters to standardize the size and concentration determination of nanomaterials by nanoparticle tracking analysis. Int J Pharm. 2024 May 10;656:124097. doi: 10.1016/j.ijpharm.2024.124097. Epub 2024 Apr 10. PMID: 38609058.
Comments 10: “10 - Include the instrumental parameters for TEM imaging (accelerating voltage, exposure time, and detector type).”
Response: The authors thank the reviewer for this technical suggestion and agree that the TEM acquisition parameters are important for reproducibility. The manuscript has been revised to include the accelerating voltage (kV), magnification range, detector type, and operating mode used during TEM imaging.
The following was added/modified;
line 218-224: Alfalfa callus culture extracts were visualized using a transmission electron microscope (JEM-1220, JEOL, Tokyo, Japan) operating at an accelerating voltage of 80 kV. Samples for TEM observations were prepared by placing droplets of the hydrocolloid suspensions onto Formvar-coated 3 mm 200 Mesh Cu grids (Agar Scientific, Stansted, UK). Immediately after air drying, the grids were examined without staining. Images were captured with an exposure time of 1.2 s, using a Morada CCD digital camera (Olympus SIS, Germany).
Comments 11: “11 - Provide higher-contrast TEM images or adjust the scale and annotations to better highlight individual nanoparticles.”
Response: The authors thank the reviewer for this useful recommendation. The TEM figures have been reprocessed to enhance contrast and scale bars and particle annotations have been added to clearly indicate nanoparticle morphology and dimensions.
The updated images have been included in the revised manuscript as Figure 4
Comments 12: “12 - Expand on the interpretation of zeta potential and PDI results, explaining their implications for colloidal stability.”
Response: The authors thank the reviewer for this insightful comment and agree that further explanation was required. The Discussion has been expanded to explain that zeta potential values more negative than −30 mV typically indicate strong colloidal stability, while values between −10 mV and −30 mV, as observed in this study, indicate moderate stability with some agglomeration tendency.
The following explanatory sentence has been added:
“The zeta potential values obtained in this study (> −13 mV) suggest moderate electrostatic stability, indicating that biomolecules in the callus extract contributed to partial nanoparticle stabilisation.”
Line 698-700: Zeta potential values more negative than −30 mV typically indicate strong electrostatic repulsion and high colloidal stability, while values between −10 mV and −30 mV correspond to moderate stability [30,53,68]. Such values suggest that the phytochemical components of the Medicago sativa callus extract, including phenolic acids, flavonoids, and proteins, adsorbed onto the nanoparticle surfaces, providing both electrostatic and steric stabilization. The relatively high PDI values (> 0.4) observed for all samples indicate a broad particle size distribution and the coexistence of small and aggregated nanoparticles, which is typical for biologically synthesized systems where multiple biomolecules act as reducing and capping agents. Similar trends have been reported for plant-mediated nanoparticle synthesis [72].
Reference: [Pochapski DJ, Carvalho Dos Santos C, Leite GW, Pulcinelli SH, Santilli CV. Zeta Potential and Colloidal Stability Predictions for Inorganic Nanoparticle Dispersions: Effects of Experimental Conditions and Electrokinetic Models on the Interpretation of Results. Langmuir. 2021 Nov 16;37(45):13379-13389. doi: 10.1021/acs.langmuir.1c02056. Epub 2021 Oct 12. PMID: 34637312.].
This revision appears in the Discussion (lines 666–675).
Comments 13: “13 - Improve the discussion concerning the absence of the typical SPR peak for AgNPs (420 nm) and explore possible causes.”
Response: The authors thank the reviewer for this important scientific comment and agree that a more detailed explanation was necessary. The Discussion has been revised to clarify why the characteristic AgNP SPR band at ~420 nm was not observed in the UV–Vis spectra. The revised manuscript now explains that strong interactions between silver nanoparticles and biomolecules from the callus extract can suppress or shift the plasmon resonance signal, particularly in green synthesis systems.
The following clarification was added:
“. However, no distinct SPR peak at ~420 nm was observed for AgNPs in this study. This may be due to overlapping absorption from plant biomolecules, NP aggregation, or the formation of very small AgNP clusters (<10 nm) that do not exhibit a defined plasmon resonance [55,68,74]. The presence of AgNPs was still confirmed by TEM and NTA, suggesting that UV–Vis alone may not be conclusive in complex biological matrices.”
“These biomolecules, mainly phenolic compounds, proteins, and flavonoids, act as both reducing and capping agents, controlling NP nucleation and growth and preventing aggregation [77,78]. This capping layer contributes to the stability and size distribution of AgNPs and AuNPs synthesised in this study.”
This revision appears in the Discussion (lines 726-731).
Comments 14: “14 - Specify the statistical software and analysis methods employed.”
Response: The authors thank the reviewer for this comment. The manuscript has been revised accordingly. It now states that all statistical analyses were performed using Microsoft Excel (Microsoft Office Professional Plus 2019). Data are expressed as mean ± standard deviation (SD), and paired two-sample t-tests were used to compare treated and control samples, while one-way ANOVA was used to analyse differences between exposure durations. A p-value < 0.05 was considered statistically significant. This information has been added to the Materials and Methods section (lines 290-298).
“Each treatment was performed in triplicate to ensure data reliability. The data were analysed using Microsoft Excel (Microsoft Office 365) and are presented as the mean ± standard deviation (SD). For each genotype, a paired two-sample t-test was performed to determine significant differences in light absorbance between control samples and those treated with AgNO₃ or HAuCl₄ at specific wavelengths. In addition, a one-way analysis of variance (ANOVA) was applied to compare differences between exposure times (24 h and 48 h). For all statistical analyses, differences were considered statistically significant at p < 0.05”
Comments 15: “15 - More clearly relate the bioactive compounds in the callus extract to the mechanism of metal ion reduction.”
Response: The authors thank the reviewer for this valuable comment and fully agree with the request. The revised manuscript now clearly explains how biomolecules present in M. sativa callus extracts contribute to the synthesis process. Specifically, phenolic compounds and flavonoids donate electrons that reduce Ag⁺ and Au³⁺ ions, while hydroxyl and carbonyl groups form complexes that stabilise the nanoparticles.
This explanation was added to the Introduction and is supported by the following sentence:
“Phenolic compounds donate electrons through hydroxyl groups, reducing Ag⁺ and Au³⁺ ions to nanoparticles, while their aromatic rings stabilise growing nanoparticle nuclei.”
Corresponding revisions were made in the Introduction (lines 132-136).
Comments 16: “16 - Avoid extrapolations regarding antibacterial or anticancer applications without experimental validation.”
Response: The authors thank the reviewer for this important remark and fully agree with the concern. Statements that could be interpreted as speculative or implying unverified biological activity have been revised. All mentions of antibacterial or anticancer activity have now been reformulated as future research directions rather than claims.
For example, the discussion was revised as follows:
“Further investigation is needed to evaluate the potential biological activity of the synthesised nanoparticles, including antibacterial effects, in future studies.”
These revisions were made in the Discussion (lines 891-892).
This wording avoids unsupported conclusions while retaining the relevance of future applications.
Comments 17: “17 - Standardize chemical symbols and formulas, ensuring consistent use of subscripts.”
Response: The authors thank the reviewer for this observation. All chemical symbols and molecular formulas (e.g., AgNO₃, HAuCl₄, H₂O₂) were checked throughout the manuscript and corrected for uniform formatting and proper subscript notation.
Corrections were applied consistently in the revised manuscript (lines 431,441)
Comments 18: “18 - Review and correct the formatting of tables.”
Response:
The authors thank the reviewer for this valuable remark. The formatting of all tables has now been revised for clarity and consistency. Table layout, numerical alignment, units, and headers have been standardised according to the journal guidelines. In addition, statistical values were reformatted to one significant figure following Reviewer 2’s recommendation, which also required restructuring the tables to present the data more clearly. Table captions and footnotes were also updated to improve readability and ensure accurate interpretation.
Updated tables are included in the revised manuscript.
Comments 19: “19 - Reduce the number of abbreviations, and define each acronym upon its first appearance.”
Response: The authors thank the reviewer for this valuable comment. The manuscript has been revised to ensure that all abbreviations are defined at their first occurrence and unnecessary abbreviations have been removed to improve readability. For example, “NPs” (nanoparticles), “TEM” (transmission electron microscopy), and “SPR” (surface plasmon resonance), EDS (Energy-Dispersive X-Ray Spectroscopy), Dynamic Light Scattering (DLS), Laser-Induced Breakdown Spectroscopy (LIBS), are now introduced clearly before use.
These changes were applied throughout the manuscript (39, 65,317,512,568, 849).
Comments 20: “20 - Conduct a final linguistic and stylistic review to enhance clarity and fluency.”
Response: The authors thank the reviewer for this final comment and confirm that a thorough linguistic revision of the manuscript has been completed. Sentence clarity, grammar, academic tone, and logical flow have all been improved while maintaining the scientific meaning. The overall readability of the manuscript has been enhanced accordingly.
The revised version reflects these improvements throughout the text.
With kind regards,
The authors
Reviewer 2 Report (New Reviewer)
Comments and Suggestions for Authors
In this study, the authors successfully demonstrated the ‘green’ biosynthesis of gold (Au) and silver (Ag) nanoparticles using extracts from callus cultures of two alfalfa (Medicago sativa L.) genotypes: 'Kometa' and 'La Bella Campagnola'. This approach is in line with current trends in nanobiotechnology, offering a sustainable, low-toxicity alternative to the chemical synthesis of nanoparticles. The authors conducted a thorough analysis of the synthesised nanoparticles using a variety of methods. This work contributes to the development of an optimised protocol for producing nanoparticles with the desired properties for subsequent biomedical and agrotechnological applications. However, there are some observations:
1. The particle sizes determined by NTA significantly exceed those determined by TEM. This discrepancy may indicate strong particle aggregation or the presence of large organic residues in the suspension. A more detailed discussion of this discrepancy is recommended.
2. The authors rightly note that the lack of additional purification (e.g. dialysis or density gradient centrifugation) could have made visualisation difficult and led to particle overlap. This comment should be moved from the 'Discussion' section to 'Materials and Methods' or the conclusion, clearly stating it as a limitation of the current protocol and recommending it as a specific improvement for future studies.
3. While the introduction lists the possible biomolecules involved in synthesis, the discussion section does not attempt to link the biochemical differences between the 'Kometa' and 'La Bella Campagnola' genotypes with the observed differences in synthesis efficiency and nanoparticle properties. The addition of a hypothesis or references to the literature would enrich the discussion.
4. The font size in the figure legends and axis labels should be increased to improve data visualisation. This should be done for Figures 5, 6 and 7.
5. 5. Statistical data should be presented with one significant figure. This aligns the accuracy of the presentation with the level of uncertainty determined by the reliability criteria used (e.g. Student's t-test or z-distribution).
Author Response
Dear Reviewer,
The authors are grateful to the reviewer for the insightful and valuable review. Please see the attached file for detailed responses to the comments!
Best regards, the Authors!
Reviewer 2 comments:
Comments and Suggestions for Authors:
In this study, the authors successfully demonstrated the ‘green’ biosynthesis of gold (Au) and silver (Ag) nanoparticles using extracts from callus cultures of two alfalfa (Medicago sativa L.) genotypes: 'Kometa' and 'La Bella Campagnola'. This approach is in line with current trends in nanobiotechnology, offering a sustainable, low-toxicity alternative to the chemical synthesis of nanoparticles. The authors conducted a thorough analysis of the synthesised nanoparticles using a variety of methods. This work contributes to the development of an optimised protocol for producing nanoparticles with the desired properties for subsequent biomedical and agrotechnological applications. However, there are some observations:
Comments 1: ‘’1. The particle sizes determined by NTA significantly exceed those determined by TEM. This discrepancy may indicate strong particle aggregation or the presence of large organic residues in the suspension. A more detailed discussion of this discrepancy is recommended.”
Response: Authors appreciate the reviewer’s valuable observation. The Discussion section has been revised to include a more detailed explanation of the observed difference between particle sizes determined by NTA and TEM. The NTA results yielded larger average particle diameters, which can be attributed to the hydrodynamic nature of NTA measurements that reflect not only the metallic nanoparticle core but also the surrounding solvation shell, adsorbed biomolecules, and possible aggregates. In contrast, TEM provides direct measurements of the solid core diameter under dry conditions. The presence of residual organic compounds from the M. sativa callus extract likely contributed to the apparent size increase detected by NTA. The revised discussion now includes this explanation and cites relevant literature
The following was added /modified: Line 769-779
‘’The particle sizes observed by TEM (5–25 nm) were lower than those detected by NTA, which also identified larger nanoparticle populations (>100 nm). This discrepancy is common in green synthesis, as NTA measures hydrodynamic particle sizes, including capping biomolecules and nanoparticle agglomerates.
Moreover, TEM provides the core diameter of nanoparticles in a dry state, while DLS and NTA measure the hydrodynamic diameter of particles dispersed in solution, which includes the solvation layer, surface-bound biomolecules from the callus extract, and occasional aggregates. These factors account for the larger mean sizes observed in DLS/NTA relative to TEM, consistent with findings in similar green synthesis systems [82].
[Tian et al. 2024].
References:
Reference: Tian Y, Tian D, Peng X, Qiu H. Critical parameters to standardize the size and concentration determination of nanomaterials by nanoparticle tracking analysis. Int J Pharm. 2024 May 10;656:124097. doi: 10.1016/j.ijpharm.2024.124097. Epub 2024 Apr 10. PMID: 38609058.
Comments 2: “2. The authors rightly note that the lack of additional purification (e.g. dialysis or density gradient centrifugation) could have made visualisation difficult and led to particle overlap. This comment should be moved from the 'Discussion' section to 'Materials and Methods' or the conclusion, clearly stating it as a limitation of the current protocol and recommending it as a specific improvement for future studies.”
Response: The authors thank the reviewer for this constructive suggestion and fully agree with the recommendation. In accordance with the comment, the statement regarding the absence of sample purification has been moved from the Discussion to the Conclusion section. In addition, it has been revised and expanded to clearly state this as a methodological limitation of the present study and to provide a specific recommendation for future work.
The revised text now reads as follows:
“However, a methodological limitation of this study is the absence of additional sample purification prior to analysis, which may have contributed to nanoparticle aggregation and partial particle overlap during imaging. To improve nanoparticle separation and enable more detailed structural characterisation, future research should incorporate purification steps such as dialysis or density gradient centrifugation.”
This revision appears in the Conclusion section (lines 971-976).
Comments 3: “3. While the introduction lists the possible biomolecules involved in synthesis, the discussion section does not attempt to link the biochemical differences between the 'Kometa' and 'La Bella Campagnola' genotypes with the observed differences in synthesis efficiency and nanoparticle properties. The addition of a hypothesis or references to the literature would enrich the discussion.”
Response: The authors thank the reviewer for this insightful comment and agree that this mechanistic link is essential. The Discussion has now been expanded to relate genotype-specific differences in synthesis efficiency to variations in phytochemical composition between ‘Kometa’ and ‘La Bella Campagnola’.
The following explanatory sentence has been added:
“These genotype-dependent differences may be associated with variations in phenolic and flavonoid content, as such biomolecules act as reducing and capping agents during nanoparticle biosynthesis, influencing nucleation efficiency.”
This addition strengthens the mechanistic interpretation and appears in the Discussion (lines 909-911).
Comments 4: “The font size in the figure legends and axis labels should be increased to improve data visualisation. This should be done for Figures 5, 6 and 7.”
Response: The authors thank the reviewer for this helpful suggestion. The font size in the figure labels and legends has been increased for Figures 5, 6 and 7 to improve readability and ensure clarity of data presentation. Updated figures have been included in the revised manuscript.
Comments 5: “5. Statistical data should be presented with one significant figure. This aligns the accuracy of the presentation with the level of uncertainty determined by the reliability criteria used (e.g. Student's t-test or z-distribution).
Response: The authors thank the reviewer for this technical clarification and agree with the recommendation. Statistical values have now been rounded to one significant figure throughout the manuscript to improve precision and consistency with statistical reporting standards. Corrections were applied to tables and figure annotations in the Results section.
With kind regards,
The authors
Round 2
Reviewer 1 Report (New Reviewer)
Comments and Suggestions for Authors
The authors implemented all the suggestions.
This manuscript is a resubmission of an earlier submission. The following is a list of the peer review reports and author responses from that submission.
Round 1
Reviewer 1 Report
Comments and Suggestions for Authors
Despite attempts to improve the manuscript, I believe that the authors have not done enough to unequivocally confirm the conclusions reached. Determining the size distribution of metal nanoparticles in systems containing some other particles (which is confirmed by the obtained results) cannot be an irrefutable confirmation of the existence of metal nanoparticles in the investigation systems. In addition, the authors showed that AuNPs and AgNPs have approximately the same concentrations in the tested samples. If this is the actual situation, it would be interesting to see an explanation for why AuNPs have a clearly defined SPR peak in their absorption spectra, while in the case of AgNPs there is no indication of an SPR peak.
Also, the authors state in conclusion that the sizes of AgNPs and AuNPs are about 17.3 nm and 57.6 nm, respectively, which differs significantly from the results of the NPs size distribution test. These results show that the mean size of AgNPs is about 220 nm, and AuNPs is about 243 nm. I believe that such differences in results deserve an explanation supported by facts.
In addition, I believe that certain references are not appropriate and are not at the appropriate position in the text. I haven't checked every single reference, but I found the following:
- Ref. 44 does not exist in the manuscript
- Ref. 50 and 51 in parts 2.1 and 2.2, respectively, have no relation to the mentioned
- Ref. 52 part 2.3 - I do not see any connection
- Ref. 19 in 2.5 is inappropriate because it is a review paper and does not deal with any experimental procedure
- Ref. 53–56 in part 2.6 - I do not understand why they are listed there
- Ref. 67-69 (lines 473-475) do not agree with the above statement. Although in the case of these references, there is no sample purification during preparation for TEM imaging, as the authors stated, all samples were purified in a previous step during the synthesis itself. The purifying step is important in these types of samples.
Author Response
Dear Reviewer,
We sincerely thank you for your valuable review. Please find attached a file in which the authors have provided detailed responses to all of your comments! Kind regards, Authors!

Reviewer 2 Report
Comments and Suggestions for Authors
In this manuscript “Using Medicago sativa L. Extract of Callus Cells for the Synthesis of Gold and Silver Nanoparticles”, the authors focused on the biosynthesis of gold (Au) and silver (Ag) nanoparticles (NPs) using extracts from Medicago sativa callus cultures. They demonstrated this strategy was successful through a series of characterizations represented by TEM, DLS, etc. This work provided a new direction for the development of plant-based nanomaterials with biomedical or environmental applications. The experimental results can support the above conclusions. Therefore, I recommend acceptance of this work in the journal. Detailed comments are shown below.
- It is mentioned that TEM images confirmed the biosynthesis of Au and Ag NPs, but the crystal structure of the NPs was not analyzed. What are their crystal structure characteristics? Supplementary characterizations including XRD, HRTEM should be conducted?
- In the experiment, the precursor concentrations were set as 200 mg/L for HAuCl4 and 1000 mg/L for AgNO3, with a significant difference between the two concentrations. What is the influence of the concentration on the synthesis of Au and Ag NPs?
- A variety of characterization methods were used in the experiment. However, could characterization methods such as XRF be further employed to analyze their purity, so as to rule out the influence of other impurities (such as unreacted precursors and other components in the extract)?
- Can the authors demonstrate the Au and Ag NPs synthesis from this work are comparable with other Au and Ag NPs in a specific application.
Author Response
Dear Reviewer,
We sincerely thank you for your valuable review. Please find attached a file in which the authors have provided detailed responses to all of your comments.
Kind regards,
The Authors

Reviewer 3 Report
Comments and Suggestions for Authors
This study demonstrated the cost-effective biosynthesis of spherical Au and Ag NPs using callus cultures of two Medicago sativa L. genotypes, ‘Kometa’ and 'La Bella Campagnola', exposed to AgNO3 and HAuCl4 for 24 and 48 hours. The synthesized NPs, confirmed by spectrophotometry, DLS, Zeta potential measurements, and TEM imaging, exhibited diverse nanoscale sizes and eco-friendly properties, making them suitable for antibacterial and anticancer applications. Overall, the manuscript is suitable for the publication in IJMS after revision. Here are some suggestions for the revision.
- The red words should be changed to black words.
- In Figures 1 and 2, the standard deviation differs between the upper and lower parts of each column. Please check the data.
- To confirm the successful synthesis of Au and Ag NPs, elemental analysis is necessary. For example, the analysis of Au and Ag NPs is recommended using energy-dispersive X-ray spectroscopy.
- Some sentences should be corrected in the manuscript. For this sentence “Due to their antibacterial, antifungal, antiviral, antiparasitic, anti-inflammatory, and anti-tumour properties.”, the sentence is a fragment because it lacks a subject and a main verb. Please check in the whole manuscript.
- In the introduction, the statement “These qualities enable both Ag and Au NPs to be used in imaging, biomedical devices, and cancer diagnostics” could be supplemented with additional literature to emphasize their importance.
https://doi.org/10.1186/s12951-023-02208-3
Author Response

(The authors gave the same response as above.)

Round 2
Reviewer 1 Report
Comments and Suggestions for Authors
Despite all attempts to improve the manuscript, I believe that the authors have not done enough to unequivocally confirm the presented conclusions. For example, EDS and LIBS are techniques that authors used to confirm the presence of AgNPs and AuNPs. But neither of these techniques do not confirms the presence of nanoparticles. These techniques confirmed the presence of the metallic state of silver and gold. As is known, silver and gold in metallic form can be present in some systems, but this does not necessarily indicate that the formation of nanoparticles has occurred. In this investigation, the presence of metallic silver and metallic gold in the systems is not questioned (ions of both metals were introduced into the system during the synthesis process), but the presence of nanoparticles in the investigated systems is questioned, especially the presence of silver nanoparticles.
Author Response
The authors thank the reviewer very much for taking the time to review this manuscript. Please find the detailed response in the attached file and the corresponding correction highlighted in the re-submitted file with comments for reviewers and the editor.

Reviewer 2 Report
Comments and Suggestions for Authors
The authors have revised the manuscript thoroughly according to the previous suggestions, and the improvements make the work more rigorous and suitable for publication. Therefore, I recommend the publication of this work
Author Response
Dear Reviewer!
The authors sincerely thank the reviewer for the positive evaluation and kind recommendation for the publication of our work. We are very grateful for the constructive feedback and suggestions provided during the review process, which helped us improve the quality and rigor of the manuscript.
The authors also wish the reviewer continued success in their valuable research and future work!
With best regards,
The Authors
Reviewer 3 Report
Comments and Suggestions for Authors
The revised has satisfied the comments from the reviewers. Therefore, the revised manuscript is now suitable for the publication in IJMS as its current form.
Author Response

(The authors gave the same response as above.)
